# DAVIDSONIAN SCENE GRAPH:
# IMPROVING RELIABILITY IN FINE-GRAINED EVALUATION FOR TEXT-TO-IMAGE GENERATION

**Jaemin Cho**[1]* **Yushi Hu**[2]  **Roopal Garg**[3]  **Peter Anderson**[3]  **Ranjay Krishna**[2]
**Jason Baldridge**[3]  **Mohit Bansal**[1]  **Jordi Pont-Tuset**[3]  **Su Wang**[3]

[1]University of North Carolina at Chapel Hill  [2]University of Washington  [3]Google Research
https://google.github.io/dsg

## ABSTRACT

Evaluating text-to-image models is notoriously difficult. A strong recent approach for assessing text-image faithfulness is based on QG/A (question generation and answering), which uses pre-trained foundational models to automatically generate a set of questions and answers from the prompt, and output images are scored based on whether these answers extracted with a visual question answering model are consistent with the prompt-based answers. This kind of evaluation is naturally dependent on the quality of the underlying QG and VQA models. We identify and address several reliability challenges in existing QG/A work: (a) QG questions should respect the prompt (avoiding hallucinations, duplications, and omissions) and (b) VQA answers should be consistent (not asserting that there is no motorcycle in an image while also claiming the motorcycle is blue). We address these issues with Davidsonian Scene Graph (DSG), an empirically grounded evaluation framework inspired by formal semantics, which is adaptable to any QG/A frameworks. DSG produces atomic and unique questions organized in dependency graphs, which (i) ensure appropriate semantic coverage and (ii) sidestep inconsistent answers. With extensive experimentation and human evaluation on a range of model configurations (LLM, VQA, and T2I), we empirically demonstrate that DSG addresses the challenges noted above. Finally, we present DSG-1k, an open-sourced evaluation benchmark that includes 1,060 prompts, covering a wide range of fine-grained semantic categories with a balanced distribution. We release the DSG-1k prompts and the corresponding DSG questions.

## 1 INTRODUCTION

Text-to-Image (T2I) generation models are the "talk of the town" (Saharia et al., 2022; Yu et al., 2022; Chang et al., 2023; Rombach et al., 2022; Ramesh et al., 2022). Until recently, the standard practice to assess **T2I alignment** was to compute similarity scores between the prompt and the generated image using captioning models (Hong et al., 2018), multimodal encoders (Xu et al., 2018; Hessel et al., 2022), or object detectors (Hinz et al., 2022; Cho et al., 2023a). To provide a more fine-grained and interpretable (hence informative) evaluation of T2I alignment, a recent line of work (Hu et al., 2023; Yarom et al., 2023; Cho et al., 2023b) proposes to use a Question Generation (QG) module to create a set of validation questions and expected answers from text description. For example, from text "*a blue motorcycle parked by paint chipped doors.*", it generates the question "*is there a motorcycle?*" and the expected answer "*Yes*". Then a Visual Question Answering (VQA) module answers the questions based on the generated image, and a score is computed by comparing the responses to the expected answers. We refer to these as **QG/A** frameworks (see Fig. 1). QG/A is well motivated by successful applications in other areas such as summarization, where QA is used to validate the quality of automatically generated summaries (Deutsch et al., 2021; Min et al., 2023).

Compared to the popular single-score similarity-based metrics (e.g., CLIPScore, Hessel et al. 2022; CLIP R-Precision, Park et al. 2021), QG/A approaches are more ***calibrated*** and ***interpretable***, and

---

*Work done as a Student Researcher at Google Research.

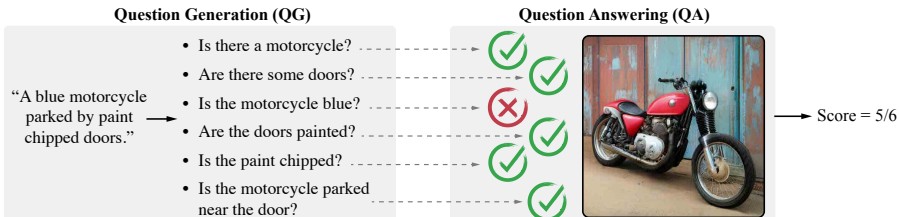

Figure 1: **QG/A workflow**: Generate questions from text prompt (QG), answer the questions via VQA models (QA), and average the answer scores to obtain a single summary score.

they **correlate well** with human judgments. For calibration, *e.g.*, a CLIPScore of 0.3 is empirically on the low side for realistic images but on the high end for pixel art. For interpretability, the question-answer pairs provides a more granular view of the model behavior. E.g., it is possible to assess that a generated image depicts the entities and their spatial relations correctly while having the colors wrong. For human correlation, Hu et al. (2023) for example achieves $47.2$ Kendall's $\tau$ in a sizable correlation analysis with human 1-5 Likert judgments, with CLIPScore $\tau = 23.1$.

Nevertheless, we identify several nontrivial reliability issues in existing QG/A methods. Extracted questions often involve multiple details that introduce ambiguity in interpretation (e.g., the answer to "*is there a blue motorcycle*" covers both the color and presence of the object). Duplicated and hallucinated questions are common (e.g., "*is there a motorcycle?*" & "*what type of vehicle is this?*" are duplicates). Finally, the VQA **invalid query failure** is prevalent (e.g., "*is there a motorcycle?*" (root), model: "*no*"; "*is the motorcycle blue?*" (dependent), model: "*yes*").

Focusing first on the QG step, we argue that a good framework should have the following properties:

A. **Atomic Questions**. Each question should cover the smallest possible semantic unit, so that (i) its answer is unequivocally interpretable and (ii) it facilitates the task of the VQA model or person answering it. E.g., "*is there a blue motorcycle?*" inquires about more than one detail and thus the answer is unnecessarily harder to assess – is there no motorcycle or is the color incorrect?

B. **Full Semantic Coverage and No Hallucinations**. All contents of the prompt, and only the contents of the prompt, should be represented by the generated questions.

C. **Unique Questions**. Having "*is there a motorcycle?*" and "*what type of vehicle is this?*" over-represents one part of the prompt and thereby skews the final metric.

D. **Valid Question Dependency**. Depending on the answers, some questions become invalid and thus should not be asked to the VQA module. For instance, if the answer to "*is there a motor-cycle?*" is no, *dependent* questions like "*is the motorcycle blue?*" should be skipped – VQA models may often say "motorcycle doesn't exist but it's blue" (more examples in Appendix C).

To capture these desired properties, we introduce **Davidsonian Scene Graph (DSG)**, a structured representation of text descriptions that naturally fulfills the above requirements. It is inspired by linguistic formalisms (Davidson, 1965; 1967a;b) that represent a sentence's semantics by decomposing it recursively down to a set of *atomic propositions* (A) that collectively represent the entirety of the semantics of the sentence, no more and no less (B). We implement the QG step as a Directed Acyclic Graph (DAG) where the nodes represent the unique questions (C) and the directed edges represent semantic dependencies between them (D). In the VQA validation step, these dependencies ensure that if the answer to a question is negative, then all the dependent questions in the sub-graph are skipped. DSG is implemented as a modular pipeline based on state-of-the-art Large Language Models – LLMs (e.g., PaLM 2 (Anil et al., 2023), GPT-3.5/4 (OpenAI, 2022; 2023)) – for the first QG stage. The second QA stage is handled by state-of-the-art VQA modules (Li et al., 2022; Chen et al., 2023; Dai et al., 2023), where the dependent questions are skipped by VQA models according to the structure in the dependency graph (Figure 2). We comparatively experiment with various LLMs and VQA modules and present our pipeline with a combination of PaLM 2 340B (Anil et al., 2023) and PaLI 17B (Chen et al., 2023). To further facilitate research in T2I alignment evaluation, we collect **DSG-1k**, a fine-grained human-annotated benchmark with a diverse set of $1,060$ prompts (Table 1) with a balanced distribution of semantic categories and styles (Fig. 4). The prompts are sourced from a wide range of existing public datasets.

Through comprehensive experimentation (Sec. 4), we found that (a) DSG addresses the aforementioned reliability issues well; (b) latest VQA models are capable to support the QG/A approach in

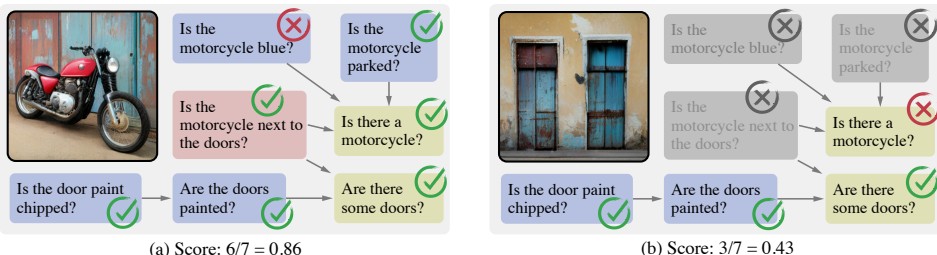

Figure 2: DSG T2I alignment evaluation. The dependencies among questions avoid invalid queries (e.g., *there isn't a motorcycle, so don't ask about its color*, in the right example).

some semantic categories but fail in others. Finally we case evaluate SoTA T2I models and thereby demonstrating DSG's strength in fine-grained diagnosis which helps advancing model development.

## 2    RELATED WORK

**Single-summary scoring frameworks.** The most common approaches for Text-to-Image (T2I) generation include text-image embedding similarity scores from multimodal encoders (*e.g.*, CLIP-Score (Hessel et al., 2022) and R-precision (Xu et al., 2018; Park et al., 2021)) and text similarity based on image captioning (Hong et al., 2018). These metrics are often uncalibrated (*e.g.*, 0.2 CLIP-Score might be high for pixel art but low for realistic photos). In addition, summarizing text and image in single embeddings obscures the semantic granularity involved in T2I alignment. SOA (Hinz et al., 2022) and DALL-Eval (Cho et al., 2023a) introduced object detection in validating text-image alignment on a finer-grained scale (and naturally, calibration by object alignment accuracy). These methods are however limited in validation scope, *e.g.*, relations among objects and global stylistic features often lie beyond the purview of object detection capabilities.

**QG/A frameworks.** Question Answering (QA)-based evaluation emerged early in text summarization to gauge the amount of key information contained in a summary: manually in Narayan et al. (2018), automatically in Durmus et al. (2020); Eyal et al. (2019), and more comprehensively in Deutsch et al. (2021). In the multimodal domain, with advancing capabilities of large pre-trained foundation models, a line of work has emerged around the idea of validating alignment by applying Visual Question Answering (VQA) on questions generated from the prompt. This approach (collectively referred to as QG/A) allows for more fine-grained evaluation (*e.g.*, How many objects are matching? Are relations and attributes correctly bound to specified objects?). TIFA (Hu et al., 2023) generates questions with GPT-3 (Brown et al., 2020) in various semantic categories (*e.g.*, color, shape, counting) and validates with VQA modules such as mPLUG (Li et al., 2022). Yarom et al. (2023) propose a similar approach with VQ$^2$A (Changpinyo et al., 2022) as the question generator and improved VQA model-human correlation through data synthesis and high quality negative sampling. VPEval (Cho et al., 2023b) further refines the process by introducing object detection and Optical Character Recognition (OCR) modules, and orchestrates the querying in the form of controllable visual programming with ChatGPT (OpenAI, 2022). Our work follows the QG/A methodology, but it takes inspiration from formal semantics (Davidson, 1965) to address several nontrivial reliability issues observed in prior work (*e.g.*, duplicated and non-atomic question).

## 3    DAVIDSONIAN SCENE GRAPH

Briefly sketched, DSG is a set of T2I alignment validation questions structured in a (directed) scene graph, produced from the prompt as the *ground truth*. The questions are posed to a pre-trained VQA module in the context of the image evaluated. The final result consists of per-question answers and an aggregated VQA accuracy score. Figure 2 illustrates the process.

**Dependency-aware atomic decomposition of scene description.** Like other QG/A-based T2I evaluation frameworks, given a text prompt $t$ and a generated image $i$, we generate a set of validation questions $\mathbf{q}$ and their expected answers $\mathbf{a}$ from $t$. We then obtain answers from VQA model $\tilde{\mathbf{a}}$ to $\mathbf{q}$ in the context of $i$. By comparing $\tilde{\mathbf{a}}$ and $\mathbf{a}$, we can compute an alignment score between $t$ and $i$: the closer $\tilde{\mathbf{a}}$ and $\mathbf{a}$ are, the more aligned the image is to the prompt.

| Entities - 40.9% | | Attributes - 23.5% | | | | | | | | Relations - 24.3% | | Global 11.3% |
|---|---|---|---|---|---|---|---|---|---|---|---|---|
| Whole | Part | State | Color | Type | Material | Count | Size | Texture | Text rend | Shape | Spatial | Scale | Global |

Figure 3: Semantic category breakdown for TIFA-160 (manual) annotation. ***Entity***: whole (entire entity, e.g., *chair*), part (part of entity, e.g., *back of chair*). ***Attribute***: color (e.g., *red book*), type (e.g., *aviator goggles*), material (e.g., *wooden chair*), count (e.g., *5 geese*), texture (e.g., *rough surface*), text rendering (e.g., letters "Macaroni"), shape (e.g., *triangle block*), size (e.g., *large fence*). ***Relation***: spatial (e.g., *A next to B*); action (*A kicks B*). ***Global*** (e.g., *bright lighting*). Appendix B presents details on per-category question counts.

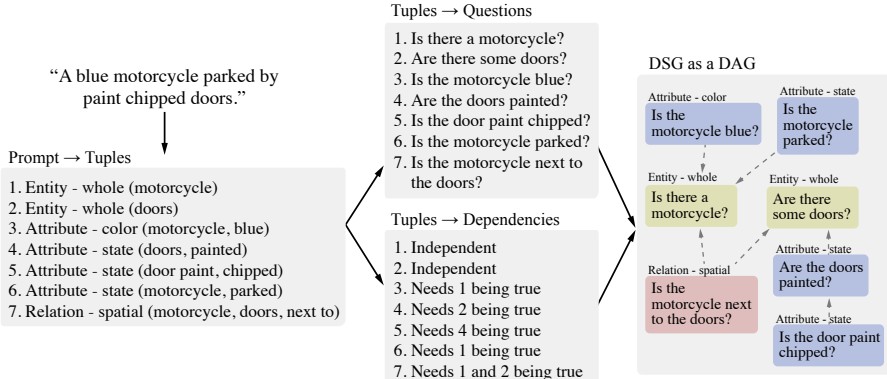

Figure 4: **Automatic generation of DSG**. We first generate semantic tuples from text prompt. Then we generate questions and dependencies from the semantic tuples, in the form of DAG.

Inspired by Davidsonian Semantics (Davidson, 1965; 1967a;b), we represent $t$'s semantics as a Directed Acyclic Graph (DAG), where the nodes are atomic propositions and edges are entailment dependencies among them. An *atomic proposition* is a statement which (i) has a *truth value* that can be verified against $i$, and (ii) cannot be further decomposed into constituent propositions with truth values. A child proposition and its parent are in an *entailment dependency* if the truth of the parent is necessary for the child to be considered. For the prompt "*a blue motorcycle*", the proposition "*there is a motorcycle*" is a parent of "*the motorcycle is blue*": this dependent's truth (i.e. whether the *motorcycle* is *blue*) can only be evaluated if the parent is true (i.e. there *is* a *motorcycle* at all).

We identify four types of atomic propositions: entities, attributes, relationships, and globals. Each type further subsumes detailed semantic categories (Fig. 3). We represent each of these with *tuples*: entities and globals are 1-tuples (the entity or global feature itself), attributes are 2-tuples (the attribute and the entity it refers to), and relationships are 3-tuples (relationship, subject entity, and object entity). Each tuple can then be translated into an atomic question that can be fed to the VQA module (Fig. 2). We call the resulting DAG Davidsonian Scene Graph (DSG).

**Automatic generation of DSGs.** We propose a three-step pipeline for automated generation of DSGs (Fig. 4): ***Prompt → Tuples***: A list of atomic semantic tuples is generated from the input prompt, each of it with a numerical identifier. ***Tuples → Questions***: Each semantic tuple is translated into a natural language question with an expected binary yes/no answer. ***Tuples → Dependencies***: The dependency between the tuples is established by generating pairs of parent-children identifiers.

Each step is implemented by an LLM given task-specific in-context examples: we prompt an LLM with a preamble (with input and output sampled from manual annotations with fixed seeds) to elicit tuple annotations of the same format for new inputs. The details on the preamble engineering is in Appendix A. This three-step pipeline proved more effective than a single-step pipeline where an LLM the whole DSG at once, since it allows us to include more task-specific in-context examples for each stage (more than 20 vs. fewer than 10).

For a generation pipeline such that the question sets resulted fulfill the four reliability properties (i.e., *atomicity, full semantic coverage, uniqueness, and valid dependency*; see Sec. 1) as faithfully as possible, we conduct two layers of quality assurance. First, we ensure that all three human

Table 1: **DSG-1k overview.** To comprehensively evaluate T2I models, DSG-1k provides 1,060 prompts covering diverse skills and writing styles sampled from different datasets.

| Feature | Source | Sample | Example |
|---|---|---|---|
| Assorted categories | TIFA160 (Hu et al., 2023) | 160 | "A Christmas tree with lights and teddy bear" |
| Paragraph-type captions | Stanford paragraphs (Krause et al., 2017) | 100 | "There is a cat in the shelf. Under the shelf are two small silver barbels. On the shelf are also DVD players and radio. Beside the shelf is a big bottle of white in a wooden case." |
| | Localized Narratives (Pont-Tuset et al., 2020) | 100 | "In this picture I can see food items on the plate, which is on the surface. At the top right corner of the image those are looking like fingers of a person." |
| Counting | CountBench (Paiss et al., 2023) | 100 | "The view of the nine leftmost moai at Ahu Tongariki on Easter Island" |
| Relations | VRD (Lu et al., 2016) | 100 | "person at table. person has face. person wear shirt. person wear shirt. chair next to table. shirt on person. person wear glasses. person hold phone" |
| Written by T2I *real* users | DiffusionDB (Wang et al., 2023) | 100 | "a painting of a huangshan, a matte painting by marc simonetti, deviantart, fantasy art, apocalypse landscape, matte painting, apocalypse art" |
| | Midjourney-prompts (Turc & Nemade, 2022) | 100 | "furry caterpillar, pupa, screaming evil face, demon, fangs, red hands, horror, 3 dimensional, delicate, sharp, lifelike, photorealistic, deformed, wet, shiny, slimy" |
| Human poses | PoseScript (Delmas et al., 2022) | 100 | "subject is squatting, torso is leaning to the left, left arm is holding up subject, right arm is straight forward, head is leaning left looking forward" |
| Commonsense-defying | Whoops (Bitton-Guetta et al., 2023) | 100 | "A man riding a jet ski through the desert" |
| Text rendering | DrawText-Creative (Liu et al., 2023) | 100 | "a painting of a landscape, with a handwritten note that says 'this painting was not painted by me'" |

experts reach consensus that each question / question set is valid for the properties for in-context examples. Second, we administer both automatic and human evaluation on the generated DSG outputs (Sec. 4.1, Tab. 2).

**DSG-1k dataset.** We construct the DSG-1k prompts by combining publicly available prompts that cover *different challenges* (e.g., counting correctly, correct color/shape/text/etc.) rendering, *semantic categories*, and *writing styles*. The core of DSG-1k is **TIFA160**, from TIFA v1.0 (Hu et al., 2023), which come with Likert-scale T2I alignment ratings by human annotators. While TIFA160 prompts come from different datasets, its size (160 prompts) limits the diversity of features, so we sample 900 more prompts from 9 different datasets, covering different writing styles (*e.g.*, short captions, paragraph-level captions, synthetic relation tuples, alt text, prompts written by real users), and skills (*e.g.*, text rendering, counting, relations), enabling evaluation of more diverse aspects of T2I generation capabilities with source-specific scores. DSG-1k has 1,060 prompts in total (Table 1). For the annotation details for DSG-1k, see Appendix B.

## 4 EXPERIMENTS AND DISCUSSION

This section covers the validation experiments we conduct in *question generation and answer* and our *T2I evaluation* results on representative SoTA T2I models. In regards to QA, we focus on pushing the limit on the direct VQA query strategy. A summary on the mixed technique involving non-VQA modules (*e.g.*, dedicated OCR module for text rendering) is in Appendix F.

### 4.1 QUESTION GENERATION

In the following, we evaluate the DSG questions in terms of how accurate they are (precision/recall), how simple/straightforward they are (atomicity), whether they are overlapping each other (uniqueness), and how accurate their inter-dependencies are.

**Do generated DSG tuples match human annotations?** *Precision & Recall.* Given a prompt, a set of human-annotated semantic tuples $T = \{t_1, \ldots, t_{|T|}\}$, and model-generated questions

| Data / Evaluator | QG Method | Precision (%) | Recall (%) |
|---|---|---|---|
| TIFA160 (30 samples) / manual | DSG (ours) | 92.2 | 100.0 |
| TIFA160 (160; full) / GPT-3.5 | | 98.3 | 96.0 |

| Data / Evaluator | QG Method | Atomicity (%) | Uniqueness (%) |
|---|---|---|---|
| TIFA160 (30 samples) / manual | TIFA | 78.5 | 65.4 |
| | VQ²A | 3.9 | 89.5 |
| | DSG (ours) | **96.5** | **97.5** |
| TIFA160 (160; full) / GPT-3.5 | TIFA | - | 73.4 |
| | VQ²A | - | 79.8 |
| | DSG (ours) | - | **90.5** |

Table 2: Evaluation of generated questions on TIFA160 prompts with manual human evaluation (on 30 prompts) and GPT-3.5 based automated evaluation (on 160 prompts). **Top**: DSG achieves high precision/recall. **Bottom**: DSG and higher atomicity and uniqueness than the baseline question generation methods.

$Q = \{q_1, \ldots, q_{|Q|}\}$; let $\mu_{t,q}$ be 1 if $q$ matches $t$ (e.g., $q$ = "*is there a motorcycle?*" matches $t$ = (entity - motorcycle)). From the tuples and questions, we can calculate ***precision*** $= \sum \mu_{t,q}/|Q|$ and ***recall*** $= \sum \mu_{t,q}/|T|$. The evaluation is done both manually and automatically. For the former, we sample 30 prompts from TIFA160 and ask 3 domain experts to provide match judgments ($\mu_{t,q}$). For the automatic evaluation, we apply an LLM (GPT-3.5 (OpenAI, 2022)) to generate question-tuple matching by showing in-context examples (see Appendix A for details).

We report results in Table 2 (top). Overall, the DSG questions are close to perfect in matching the source semantic tuples, in both manual (precision 92.2% / recall 100%) and automatic evaluations (precision 98.3% / recall 96.0%). On examining manual annotations, we observe three types of mismatches: ***Granularity of relation decomposition***, e.g.,, in "*A table topped with bags of luggage and purses.*", the human annotation holds the relationship between the *purses* and *table* as implicit, whereas DSG includes the question explicitly. ***Granularity in referential scope***, e.g.,, in "*[...] two wheels [...] one behind the other*", the human annotation considers the wheels a single entity, whereas DSG generates questions both about front and back wheels. ***Subjective details***, e.g.,, in "*A group of giraffes are gathered together in their enclosure.*", the possessive relationship *the enclosure is the giraffes'* isn't included in the human annotation whereas DSG produces "*is the enclosure theirs (the giraffes)?*". The matter with granularity is relatively minor but subjectivity deserves a dedicated solution as a valid QG/A method should distinguish between the details that can/cannot be visually verified and exclude the latter.

**Are the questions simple and not overlapping each other?** *Atomicity & Uniqueness.* A question is atomic if it only asks about one fact (an entity/attribute/relation), *e.g.*, "*is this a motorcycle?*" is atomic whereas "*is this a blue motorcycle?*" is not (asks about the entity "*motorcycle*" and its color attribute "*blue*"). We ask two experts to make a judgment on the same 30 prompt samples above, and calculate the percentage of atomic questions for TIFA, VQ²A, and DSG. We observe that LLMs do not yield reliable answers to this concept yet. We define uniqueness as the proportion of unique questions in a question set. For example, in $Q = \{$"*is there a motorcycle?*", "*what type of vehicle is this?*"$\}$, the questions are duplicates, thus uniqueness is $1/2 = 0.5$.

We report results in Table 2 (bottom) both for manual evaluation on 30 samples (3 expert annotators) and automatic one on all of TIFA160. Overall, DSG achieves strong performance in keeping questions atomic (easier to obtain unambiguous answers) and unique (thereby avoiding skewing metric calculation). For atomicity, we observe a large number of non-atomic questions in VQ²A, leading to very low atomicity of (3.9%), *e.g.*, "*A human is right to what?*" points to all entities to the right of the person. This frequently causes the VQA model to give a valid answer which happens to not match the ground truth. For uniqueness, VQ²A and DSG fare relatively better. Figure 9 in appendix exemplifies disagreement between auto and human evals.

**Are the question dependencies valid?** DSG produces parent-child annotations where the child question is only valid (as a VQA query) if the answer to the parent question is positive (*e.g.*, "*is the motorcycle blue?*" is only valid if the answer to "*is there a motorcycle?*" is positive). We formulate the evaluation as binary judgments for all parent-child pairs, and conduct it manually on 30 samples (3 expert annotators) and automatically on the full TIFA160. In the automatic evaluation, we check whether the entity/attribute/relation mentioned in the parent is also included in the child. ***On the 30 samples from TIFA160, the valid ratio is 100%. For the full TIFA160, barring a few failure cases with parsing errors, the ratio remains close to perfect, at 99%.*** Given the qualitative benefit of dependency annotations in avoiding incurring invalid VQA answers, this result strongly supports the reliability of DSG in this regard.

| VQA models | QG methods | | |
|---|---|---|---|
| | TIFA | VQ$^2$A | DSG |
| mPLUG-large | 0.352/0.259 | 0.212/0.164 | 0.463/0.380 |
| Instruct-BLIP | 0.460/0.360 | 0.426/0.352 | 0.442/0.364 |
| PaLI | 0.431/0.323 | 0.207/0.157 | **0.571/0.458** |

Table 3: Per-item evaluation of VQA models combined with different QG methods. We compare Spearman's $\rho$ and Kendall's $\tau$ correlation between VQA score and human 1-5 Likert scores on TIFA160 prompts. The **correlation** section in Appendix E further contextualizes the results with single-summary metric (*e.g.*, CLIPScore) results.

| VQA models | DSG-1k Question Types | | | | |
|---|---|---|---|---|---|
| | Entity | Attribute | Relation | Global | Overall |
| mPLUG-large | 81.4 | 61.3 | 74.7 | 45.6 | 72.1 |
| Instruct-BLIP | 79.6 | 59.5 | 72.0 | **48.3** | 70.4 |
| PaLI | **83.0** | **62.3** | **77.2** | 47.2 | **73.8** |

Table 4: VQA-Human Answer ***match accuracy*** on three T2I generation models on DSG-1k prompts, averaged over questions. We use DSG-PaLM2 questions.

## 4.2 QUESTION ANSWERING

With questions properly generated, ***a good VQA module is expected to answer the questions in high alignment to human answers***. Following previous work (Hu et al., 2023; Yarom et al., 2023; Cho et al., 2023b), we first investigate the correlation between VQA accuracy and Likert human ratings (1-5) of text-image pairs. In addition, we examine per-question answers from VQA modules vs. human raters. We collect per-item (text-image pair) for TIFA160 prompts, and per-question human judgments for the full DSG-1k prompts. A per-item judgment is a 1-5 Likert text-image consistency scale, and a per-question judgment is a binary (DSG and VQ$^2$A questions) or multi-choice (TIFA questions) answers. For per-item judgment, following Hu et al. (2023), we use the five text-to-image generation models: three versions of Stable Diffusion (SD) v1.1/v1.5/v2.1 (Rombach et al., 2022), minDALL-E (Kim et al., 2021), and VQ-diffusion (Gu et al., 2022) on TIFA160 prompts. For per-question judgment, we collect three answers per questions on three text-to-image generation models: Stable Diffusion v2.1, Imagen* (Saharia et al., 2022), and MUSE* (Chang et al., 2023) (see Sec. 4.3 for details of Imagen* and MUSE*). See Appendix H for the annotation UI. Leveraging the rich annotations collected, we conduct both ***per-item*** (VQA-human Likert score correlation) and ***per-question*** evaluation (VQA-human matching accuracy). For specific VQA modules, we experiment with three state-of-the-art models PaLI (Chen et al., 2023), mPLUG-large (Ye et al., 2023), and Instruct-BLIP (Dai et al., 2023). In addition, we present supporting evidence for the value of DSG dependency annotations (as presented here) in an ablation study in Appendix D.1.

**Per-item eval.** In Table 3, we report the correlation between the average VQA scores and human (1-5 Likert) ratings on per-item text-image alignment, measured with Spearman's $\rho$ and Kendall's $\tau$. We find that DSG+PaLI combination achieves the best correlation ($\rho = 0.563$, $\tau = 0.458$) which is also strong on the absolute scale (conventionally interpreted as "*moderate to strong*" correlation). As DSG questions achieve the highest correlations among all QG methods, we use the DSG questions for the per-question evaluation of VQA described below.

**Per-question eval.** Table 4 summarizes the *proportion of matching questions* between VQA models and human raters. Overall, PaLI performs the best with 73.8% matching ratio, slightly higher than mPLUG-large and Instruct-BLIP. Breaking down by broad semantic categories (i.e., entity/attribute/relation/global), the all three VQA models output more accurate answers to entity questions than attributes/relations/global questions.

Table 5 further looks at the detailed semantic categories. For ***entities***, the matching ratio is high (and at similar level) for both whole (*e.g.*, cat) and part (*e.g.*, *neck of a cat*). Similar observation is made for ***relations*** between spatial and action. The metric distributes unevenly for ***attributes***: visually explicit categories such as color, texture, etc. are relatively much easier to handle than abstract / higher-order ones like count and text rendering. For text rendering in particular, the model-human matching ratios are (well) below 40% (the "random" ratio for the multichoice questions which are primarily binary). The observation, we believe, supports the view that training data scaling for recent large VQA models *does not* translate well into helping them learn "deep semantics" which are visually more implicit (*e.g.*, PaLI does not handle text rendering better than

Table 5: VQA-Human answer *match accuracy* on images from T2I generation models on DSG-1k prompts, **by fine-grained semantic categories**. We use DSG-PaLM2 questions and images generated by 3 T2I models: SDv2.1, Imagen*, and MUSE*. *ent*: entity, *rel*: relation, *att*: attribute.

| VQA models | DSG-1k Question types | | | | | | | |
|---|---|---|---|---|---|---|---|---|
| | (ent) whole | (rel) spatial | (att) state | (ent) part | (global) | (att) color | (rel) action | (att) type |
| mPLUG-large | 80.9 | 75.4 | 60.5 | 83.8 | 45.6 | 81.4 | 70.2 | 52.6 |
| Instruct-BLIP | 78.5 | 72.3 | 60.7 | 84.3 | **48.3** | 78.8 | 70.1 | **53.7** |
| PaLI | **82.6** | **78.3** | **62.8** | **84.6** | 47.2 | **83.6** | **70.3** | 55.5 |
| | (att) count | (att) text rendering | (att) material | (att) shape | (att) style | (att) texture | (att) size | - |
| mPLUG-large | 67.2 | **24.9** | 56.9 | **54.6** | **23.6** | 69.4 | **75.2** | - |
| Instruct-BLIP | 63.6 | 14.1 | 53.3 | 53.0 | 21.1 | 64.7 | 67.6 | - |
| PaLI | **67.4** | 13.7 | **59.1** | 53.9 | 21.1 | **70.4** | 72.4 | - |

*Subjectivity*

*Domain knowledge*

Figure 5: Questions involving *subjectivity* or *domain knowledge*. For subjectivity, in addition to the frequent VQA-human disagreement, consensus among human raters is hard to achieve. e.g,., how detailed should the image be to count as "*highly detailed*"? For domain knowledge, depending on their knowledge, average human raters may find it difficult to give reliable judgments for this type of questions. *e.g.*, what counts as "*breath of the wild*" or "*Studio Ghibli*" styles?

mPLUG-large despite the size advantage and superior performance in visual explicit categories such as `color` or `shape`.). Table 5 also reveals the difficulties of VQA models in dealing with questions in categories like `material`, `shape`, `style`, etc. Manually examining model predictions, we believe the issue primarily lies with a) *subjectivity* and b) *domain knowledge*. For subjectivity, *e.g.*, human raters often disagree with VQA on if an objective is "*made of alloy*" (`material`), whether it is "*roundish*" (`shape`), or whether a painting is "*highly detailed*" (`style`); and they also disagree among themselves. For domain knowledge, it is generally impractical to assume the average human rater possesses the knowledge required to make an objective judgment. Fig. 5 exemplifies.

## 4.3 TEXT-TO-IMAGE EVALUATION

We compare three recent T2I generation models: SD v2.1 (Rombach et al., 2022), Imagen* (Saharia et al., 2022), and MUSE* (Chang et al., 2023),[1] by measuring the average VQA scores calculated by our DSG. In addition to the global average scores, we calculate the scores on subsets of prompts (Table 1) of DSG-1k to understand how prompt styles affect the performance of the models. In particular, through this experiment, we intend to understand the extent to which the QG/A approach is reliable in automating T2I alignment evaluation. For the evaluation of T2I generation models, we use the best QG+QA combination from the evaluations described above: DSG questions + PaLI.

Tab. 6 summarizes our results. Overall, PaLI predictions match human ratings in ranking model performance as **Imagen* ≃ MUSE* > SD v2.1**. The score scaling however shows PaLI is overly optimistic ($\sim 80\%$ accuracy) per human ratings ($\sim 60\%$ accuracy). Observing the results per data sources, PaLI ratings track humans' closely in the categories [`TIFA160`, `Paragraph`, `Relation`,

---

[1]Imagen* and MUSE* are model variants based on the Imagen (Saharia et al., 2022) and Muse (Chang et al., 2023) models, but parameterized differently and trained on internal data sources.

Table 6: Comparison of three T2I models: SD v2.1 vs. Imagen* vs. MUSE* in VQA scores *by different DSG-1k prompt sources*. QG: DSG-PaLM2; QA: both PaLI and human raters.

| T2I models | DSG-1k Prompt Sources | | | | | | | | |
|---|---|---|---|---|---|---|---|---|---|
| | TIFA-160 | Paragraph | Relation | Counting | Real users | Poses | Commonsense-defying | Text | Overall |
| **Answerer: PaLI** | | | | | | | | | |
| SD v2.1 | 88.1 | 85.1 | 34.4 | 70.4 | 89.7 | 89.6 | 84.4 | 83.7 | 80.5 |
| Imagen* | **95.1** | **92.1** | **40.6** | 72.1 | **91.1** | 89.3 | 82.8 | 86.0 | **83.9** |
| MUSE* | 92.5 | 91.8 | 39.5 | **72.5** | 89.7 | **90.2** | **84.9** | **86.5** | 83.4 |
| **Answerer: Human** | | | | | | | | | |
| SD | 80.1 | 74.9 | 27.2 | **48.0** | 46.9 | 59.5 | 69.6 | 50.8 | 59.1 |
| Imagen* | **88.0** | **86.2** | **32.8** | 47.0 | **49.4** | **73.1** | **79.6** | **56.4** | **66.2** |
| MUSE* | 86.3 | 85.5 | 31.2 | 45.2 | 44.2 | 69.4 | 73.4 | 52.8 | 63.1 |

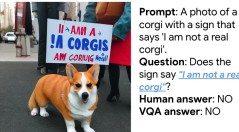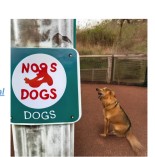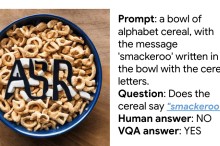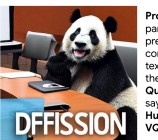

Figure 6: Judgment matching between VQA (PaLI here) and human raters in `text` (text rendering) questions. Example 1: Match, only account for ∼30% of the cases. Example 2: Mismatch, pragmatics – humans interpret the question as "*does the sign ONLY say 'no dogs'?*". Example 3 & 4: Mismatch, VQA almost invariably answer YES as long as *some text* is present.

`Commonsense-defying]` (the difference is explainable with PaLI's good yet imperfect VQA accuracy (Tab. 4, at 73.8%)), while being considerably higher in categories [`Counting, Real users, Text`]. For `Counting`, even the latest VQA models struggle to count reliably to 5: we found PaLI is incorrect in ∼ 40% of the cases (corroborated by recent research e.g., Paiss et al. (2023)). For `Text`, both T2I and VQA models fail the majority of the time (Fig. 6 exemplifies). In general, VQA performance is much lower in more ***abstract / semantically higher-order categories*** like these (Tab. 16 presents further evidence with accuracy by fine-grained categories). For `Real users`, our findings coincide with Sec. 4.2: ***Subjectivity.*** Many prompts involve details such as "*is the cyberpunk city highly detailed?*", "*is the art elegant?*" which lead to both VQA-human disagreement as well as disagreement among human raters. ***Domain knowledge.*** Some semantic elements are not reliably judgeable even by human raters, e.g., "*is this image 8k*", etc. Fig. 5 exemplifies. Based on the results, we believe a) for model-level eval on semantically more concrete categories, current VQA models are capable to support QG/A approach, yet b) at the level of particular semantic details (in particular the "difficult categories" shown above), the VQA predictions are yet sufficiently reliable.

## 5   CONCLUSION

The QG/A framework (Hu et al., 2023; Yarom et al., 2023; Cho et al., 2023b) leverages the power of pretrained LLMs and VQAs to have enabled much more fine-grained (thus informative) diagnostics in T2I evaluation. However, they have various reliability issues, such as duplicated, invalid, and ambiguous questions. In this work, inspired by formal semantics (Davidson, 1965; 1967a;b), we propose **Davidsonian Scene Graph (DSG)**, a QG/A approach to present a solution. Comprehensive experimentation and human evaluation demonstrate that DSG better addresses the reliability issues than previous work. In addition, its fine-grained question typing allows for even more in-depth diagnostics. Finally, we collect a benchmark dataset **DSG-1k** to facilitate research in this area. Our work casts light on future directions in the QG/A framework: In both QG and QA, we observe that (a) subjectivity and domain knowledge may lead to model-human as well as human-human disagreement, and (b) some semantic categories remain beyond the capabilities of existing SoTA VQA models (*e.g.*, `text rendering`).

# 6 STATEMENTS

## 6.1 ETHICS STATEMENT

Our work involves human evaluation where human raters are hired from a Google's crowdsourcing platform, and asked to look at images and and text prompts then answer questions regarding text-image grounding. This work was carried out by participants who are paid contractors. Those contractors received a standard contracted wage, which complies with living wage laws in their country of employment. Due to global privacy concerns, we cannot include more details about our participants, e.g., estimated hourly wage or total amount spent on compensation.

## 6.2 REPRODUCIBILITY STATEMENT

All the experiments involving publicly available packages and tools are reproducible (the main text and appendix provide necessary details). This includes

- GPT-3.5 (OpenAI, 2022) based precision, recall, and uniqueness evaluation described in Tab. 2;
- VQA query results in Tab. 4, and Tab. 5 with mPLUG-large (Ye et al., 2023) and Instruct-BLIP (Dai et al., 2023);
- Stable Diffusion (Rombach et al., 2022) produced images used in all parts of Sec. 4.3.

The other LLMs and VQAs used, specifically PaLM2 (Anil et al., 2023) and PaLI (Chen et al., 2023) are not yet released. Further, the human evaluation results may not be exactly reproduced due to the use of different human raters, however the general trend should retain to the extent that the conclusions drawn in this work should not be impacted.

## ACKNOWLEDGMENTS

We would like to thank Yonatan Bitton, Dennis Shtatnov, Shuai Tang, Dilip Krisnan for their generous help through the course of the project. We give thanks to Muhammad Adil, Aditya Saraswat, Sudhindra Kopalle, Yiwen Luo and all the anonymous human annotators for assisting us to coordinate and complete the human evaluation tasks. We are grateful to Arjun Akula and Radu Soricut for reviewing the paper and providing feedback. Thanks to Jason Baldridge for help with resource coordination.

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

# Appendix

In this appendix, we include LLM preambles (Appendix A), DSG-1k dataset details (Appendix B), invalid VQA query examples (Appendix C), DSG abalation studies (Appendix D), additional QG/QA/T2I evaluation results (Appendix E), experiments on mixed QA evaluation models (Appendix F) Pseudocode of T2I evaluation with DSG (Appendix G), and human evaluation setup (Appendix H).

## A  LLM PREAMBLES

Fig. 7 presents the preambles for the three-stage automatic DSG generation pipeline, with PaLM2 (Anil et al., 2023).

Fig. 8 presents the preambles for question evaluation in *precision*, *recall*, and *uniqueness*, with GPT-3.5 (OpenAI, 2022).

---

*Tuple generation*
```
Task: given input prompts, describe each scene with skill-specific tuples.  Do not
generate same tuples again. Do not generate tuples that are not explicitly described in
the prompts.
output format: id | tuple
${HUMAN_ANNOTATION_EXAMPLES}
```

*Question generation*
```
Task: given input prompts and skill-specific tuples, re-write tuple each in natural
language question.
output format: id | question
${HUMAN_ANNOTATION_EXAMPLES}
```

*Dependency generation*
```
Task: given input prompts and tuples, describe the parent tuples of each tuple.
output format: id | dependencies (comma separated)
${HUMAN_ANNOTATION_EXAMPLES}
```

---

Figure 7: The preambles for the three-step automatic DSG generation pipeline.

---

*Precision*
```
Task: a model generated questions to check semantics of images generated from prompts.
Given prompts (ground-truth), tuples that decompose the prompts (ground-truth), and
questions generated by the model, check the ids of the questions that are not entailed
with the tuples. entailed questions: id of questions | ids of tuples entailed with each
question wrong questions: ids of questions that are not entailed with GT tuples.
```

*Recall*
```
Task: a model generated questions to check semantics of images generated from prompts.
Given prompts (ground-truth), tuples that decompose the prompts (ground-truth), and
questions generated by the model, check the ids of tuples are covered by the questions.
covered tuples: id of tuples | ids of questions covering each tuple missed tuples: ids of
missed tuples
```

*Uniqueness*
```
Task: a model generated questions to check semantics of images generated from prompts.
Given prompts (ground-truth), and questions generated by the model, check the ids of
duplicated questions. output format: ids of duplicated questions
```

---

Figure 8: The preambles for the automatic evaluation of questions in precision/recall/uniqueness.

## B  DSG-1K DATASET DETAILS

**Images.** We generate images from the prompts in DSG-1k using three recent SoTA text-to-image models (SoTA at the time of experimentation): Imagen* (Saharia et al., 2022), MUSE* (Chang et al., 2023), and SD v2.1 (Rombach et al., 2022). Imagen* and SD are denoisng diffusion models (Sohl-Dickstein et al., 2015; Ho et al., 2020). MUSE* is an encoder-decoder model with parallel image decoding steps. In total, we provide $3 \times 1,060 = 3,180$ model-generated images. For fair comparison with uniform model evaluation and human annotation, we also make use of the generated images published by TIFA for TIFA160. This includes SD v1.1, v1.5 (Rombach et al., 2022), minDALL-E (Kim et al., 2021), and VQ-diffusion (Gu et al., 2022).

**Questions, Answers and Alignment Scores.** On TIFA160 prompts, a set of three human experts manually and independently generate the semantic tuples and their dependencies (scene graph). Then, with the LLM-based DSG generation pipeline (Sec. 3), we generate DSG (semantic tuples, questions, and dependencies) on the entire DSG-1k prompts. We show the generated questions together with the generated images to human annotators to (i) annotate their validity (*e.g.*, it is not valid to ask whether the motorbike is red if there is no motorbike in the image) and (ii) answer the valid questions with Yes or No. We also answer the same questions with automatic VQA models, specifically mPLUG-large (Ye et al., 2023), Instruct-BLIP (Dai et al., 2023), and PaLI (Chen et al., 2023). Following TIFA, we ask human annotators to collect 1-to-5 Likert-scale scores about T2I alignment on TIFA160 for seven models, for a total of $7 \times 160 = 1,120$ text-image pairs. In experiments, we use human Likert annotations to measure the correlation between human and QG/A judgments.

**Additional Statistics.** The #questions tables below complement Fig. 3 on the statistics for DSG-1k. In particular, the counts demonstrate the by-category analyses are valid in size statistically.

| DSG-1k | entity | attribute | relation | global |
|---|---|---|---|---|
| # questions | 3,398 | 2,251 | 2,082 | 527 |

Table 7: DSG-1k statistics: The number of questions per ***broad semantic category***.

| | ent-whole | ent-part | att-state | att-color | att-type | att-material | att-count |
|---|---|---|---|---|---|---|---|
| # questions | 2,751 | 647 | 930 | 427 | 244 | 98 | 190 |
| | att-size | att-texture | att-text rendering | att-shape | rel-spatial | rel-scale | global |
| # questions | 85 | 79 | 110 | 88 | 1,808 | 274 | 527 |

Table 8: DSG-1k statistics: The number of questions per ***detailed semantic category***. *ent*: entity, *rel*: relation, *att*: attribute.

```
Text: A chair in the corner on a boat.
Questions generated by TIFA
q1 | is there a chair in the corner?
q2 | is this a boat?
q3 | what type of vehicle is this?
q4 | is the chair in the corner?
q5 | is the chair on the boat?
GPT-3.5 eval:
Questions 1 and 4 are duplicates because they both ask about the
presence and location of the chair in the corner. duplicates:
q1,q4
uniqueness = n_unique / n_total = 4/5 = 0.8
Human eval:
duplicates: (q1,q4), (q2, q3)
uniqueness = n_unique / n_total = 3/5 = 0.6
```

```
Text: an ostrich standing on a couch.
Questions generated by DSG
q1 | Is there an ostrich?
q2 | Is there a couch?
q3 | Is the ostrich standing?
q4 | Is the ostrich on the couch?
GPT-3.5 eval:
Questions 3 and 4 are duplicates because they both ask about
the ostrich's position. duplicates: q3,q4
uniqueness = n_unique / n_total = 3/4 = 0.75
Human eval:
duplicates: None
uniqueness = n_unique / n_total = 4/4 = 1.0
```

Figure 9: Automatic uniqueness evaluation: despite the general high agreement between GPT-3.5 and human, GPT-3.5 is sometimes distracted by text overlap to err. Left: duplicates q2 & 3 differ in wording. Right: q3 & 4 are similar in wording but are unique questions.

## C  INVALID VQA QUERY EXAMPLES

Fig. 10 provide examples of invalid VQA queries described in Sec. 1. In the bottom right image, VQA model answered "*No*" to the question "*is there a car?*", but the VQA model answered "*Yes*" to the question "*is the car playing soccer?*". Such examples are more common when T2I model miss entities (more frequent in weaker models we evaluated, such as minDALL-E and VQ-Diffusion).

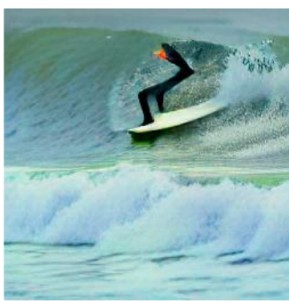

**Prompt**: A man in a wet suit is surfing.

**Root**: Is there a man?
**VQA answer**: NO

**Dependent**: Is the man surfing?
**VQA answer**: YES

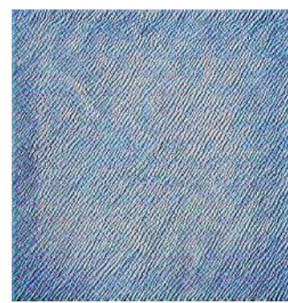

**Prompt**: A cube made of denim. A cube with the texture of denim.

**Root**: Is there a cube?
**VQA answer**: NO

**Dependent**: Is the cub made of denim?
**VQA answer**: YES

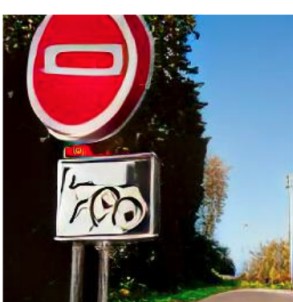

**Prompt**: a photo of bike and stop sign; stop sign is below bike.

**Root**: Is there a bike?
**VQA answer**: NO

**Dependent**: Is the stop sign below the bike?
**VQA answer**: YES

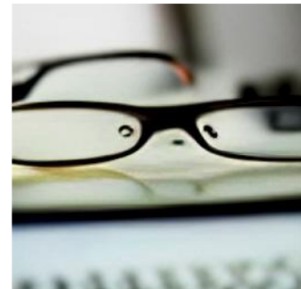

**Prompt**: a pair of glasses under a computer monitor.

**Root**: Is there a computer monitor?
**VQA answer**: NO

**Dependent**: Is the pair of glasses under the computer monitor?
**VQA answer**: YES

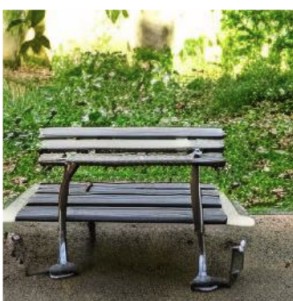

**Prompt**: a photo of suitcase and bench; bench is left to suitcase

**Root**: Is there a suitcase?
**VQA answer**: NO

**Dependent**: Is the bench left to the suitcase?
**VQA answer**: YES

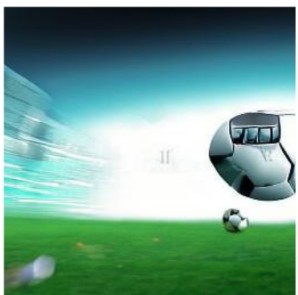

**Prompt**: A car playing soccer, digital art.

**Root**: Is there a car?
**VQA answer**: NO

**Dependent**: Is the car playing soccer?
**VQA answer**: YES

Figure 10: Examples of ***invalid VQA queries*** where a dependent question gets positive answer when its root question gets negative. For example, in the bottom right image, VQA model answered "*No*" to the question "*is there a car?*", but the VQA model answered "*Yes*" to the question "*is the car playing soccer?*". Such examples are more common when T2I model miss entities (more frequent in weaker models we evaluated, such as minDALL-E and VQ-Diffusion). Fundamentally however this is an issue of posing invalid queries to VQA that should not be posed in the first place.

## D  DSG ABLATION

### D.1  WAYS TO UTILIZE DEPENDENCY

We ablate on three ways in which DSG questions are administrated to VQA modules based on question dependencies.

1. ***DSG***. This is the method we present in the main text – if the answer to a parent question is NO, all the answers to its child questions are counted as NO as well (*e.g.*, "*is there a bike?*" NO → "*is the bike blue?*" := NO).
2. ***DSG (drop)***. If the answer to a parent question is NO, drop its child questions by not counting their answers towards the calculaton of the VQA accuracy for the text-image pair item.
3. ***DSG (w/o dep)***. Disregarding the dependency annotation and factoring in the answers to all questions equally in VQA accuracy calculation.

In gist, we found

   A. In relation to the correlation between VQA accuracy (auto) and 1-5 Likert judgments (human), the different ways to handle dependency does not lead to significant effect. Tab. 9.

   B. In the per-question VQA-human matching accuracy, on the other hand, utilizing dependencies (with either **DSG or DSG (drop)** improves performance substantially. Tab. 10.

Based on the observation, we argue that A as an aggregated reflection on the general correlational trend does not tell the whole story – B on the other hand indicates that human raters do implicitly take the question dependency into consideration, which highlights the value of incorporating dependencies in improving the reliability and interpretability of automatic evaluation results on the fine-grained per-item/question level.

| VQA models | DSG | DSG (drop) | DSG (w/o dep) |
|---|---|---|---|
| mPLUG-large | 0.463/0.380 | 0.462/0.370 | 0.464/0.378 |
| Instruct-BLIP | 0.442/0.363 | 0.437/0.358 | 0.436/0.355 |
| PaLI | **0.571/0.458** | 0.561/0.451 | 0.570/0.457 |

Table 9: DSG ablation wrt. the correlation (Spearman's $\rho$ / Kendall's $\tau$) between VQA accuracy and human 1-5 Likert score.

| VQA models | DSG | DSG (drop) | DSG (w/o dep) |
|---|---|---|---|
| mPLUG-large | 72.1 | 69.9 | 64.7 |
| Instruct-BLIP | 70.4 | 68.4 | 64.0 |
| PaLI | **73.8** | 71.4 | 65.2 |

Table 10: DSG ablation wrt. the per-question VQA-human matching accuracy.

### D.2 BY-CATEGORY ABLATION FOR TIFA, VQ$^2$A, AND DSG

TIFA160 comes with 4 data splits:

(a) **COCO**. General purpose;
(b) **PaintSkill**. Focuses on spatial relations.
(c) **PartiPrompts**. Focuses on attributional failure cases observed (per Yu et al. (2022));
(d) **DrawBench**. Focuses on compositionality.

Leveraging the full human annotations on TIFA160, we ablate across TIFA, VQ$^2$A, and DSG on the data splits in the correlation between VQA results and human Likert judgments, as a further breakdown and extension for Tab. 3. Tab. 11 summarizes the results.

| | COCO | PaintSkill | PartiPrompts | DrawBench | Overall |
|---|---|---|---|---|---|
| TIFA | 0.30/0.22 | 0.56/0.40 | 0.49/0.37 | **0.59/0.46** | 0.43/0.32 |
| VQ$^2$A | 0.17/0.13 | 0.25/0.17 | 0.28/0.21 | 0.45/0.37 | 0.21/0.16 |
| DSG | **0.53/0.42** | **0.64/0.53** | **0.61/0.49** | 0.56/0.44 | **0.57/0.46** |

Table 11: Ablation by data split for TIFA160 across QG/A methods on the correlation between VQA score and human 1-5 Likert score (Spearman's $\rho$ / Kendall's $\tau$). The VQA model used: PaLI.

## E ADDITIONAL QG/QA/T2I EVALUATION RESULTS

**Single-summary metric context.** To contextualize the correlation results between VQA scores and human 1-5 Likerts (Tab. 3) in a wider context with single-summary metrics (*e.g.*, CLIPScore), we present the correlation results between the single-summary scores investigated in TIFA (Hu et al., 2023) with the human scores we collected.[2] Following Hu et al. (2023), we report 5 single-summary

---

[2]Note: The results reported in the Table 2 in Hu et al. (2023) are not directly applicable – as the human scores are collected with different human raters.

metrics: BLEU-4, ROUGE-L, METEOR, SPICE, (with BLIP-2 Captioning) and CLIPScore (with CLIP ViT-B/32), in Tab. 12. While some single-summary correlations are higher than TIFA / VQ$^2$A in some configurations, DSG, on the other hand, results in much higher correlations with all VQA module (see Tab. 3 for details).

| Metric | BLEU-4 | ROUGE-L | METEOR | SPICE | CLIPScore |
|---|---|---|---|---|---|
| $\rho/\tau$ | 0.261/0.183 | 0.342/0.239 | 0.379/0.269 | 0.329/0.236 | 0.276/0.191 |

Table 12: Spearman ($\rho$) / Kendall ($\tau$) correlation between single-summary metrics and human 1-5 Likerts on TIFA160 prompts (across the same 5 models per Hu et al. (2023)).

**QG evaluation.** In Fig. 9, we show some example mismatch case of human evaluation and GPT-3.5 based automatic evaluation on uniqueness, described in Sec. 4.1.

**QA evaluation.** In Table 13, we also summarize the ***VQA-human match accuracy by DSG-1k data sources***. The trend matches the observation in the by-category results in Table 4 and 5.

| VQA models | DSG-1k Prompt Sources | | | | | | | |
|---|---|---|---|---|---|---|---|---|
| | TIFA-160 | Paragraph | Relation | Counting | Real users | Poses | Commonsense-defying | Text |
| mPLUG-large | 86.4 | 84.7 | 80.4 | 65.7 | 46.1 | 69.0 | 80.4 | **60.3** |
| Instruct-BLIP | 86.3 | 83.6 | 71.1 | 60.2 | 47.3 | **69.9** | 81.0 | 57.0 |
| PaLI | **87.9** | **86.4** | **84.9** | **66.1** | **47.6** | 69.7 | **82.4** | **60.3** |

Table 13: VQA-Human answer ***match accuracy*** on 3 T2I generation models on DSG-1k prompts, by data sources/segments. We use DSG-PaLM2 questions.

| minDALL-E | VQ-Diffusion | SD v1.1 | SD v1.5 | SD v2.1 | Imagen* |
|---|---|---|---|---|---|
| 3.839 | 3.608 | 3.731 | 3.990 | 4.146 | **4.399** |

Table 14: Avg. Likert (1-5) on T2I generation models on TIFA160 prompts.

**T2I evaluation.** Table 14 shows the average Likert (1-5) scores of T2I generation models on TIFA160 prompts. Table 15 and Table 16 present VQA-human correlation and accuracy results by fine-grained semantic categories. In Table 15, we calculate for categories with $\geq 30$ samples and keep results with $p < 1e - 4$.

## F    EXPERIMENTS ON MIXED QA EVALUATION MODELS

In the QG/A framework, in relation to question answering specifically, an alternative strategy has been explored in concurrent work. One prominent representative is VPEval (Cho et al., 2023b), where non-VQA modules are applied in corresponding semantic categories – specifically a) object detection (OD) for `counting`, and b) optical character recognition (OCR) for `text rendering`.

Concretely, for evaluation queries measuring counting skill (*e.g.*, "Are there five Halloween banners?"), an OD module detect bounding boxes with the target object name mentioned in the question. The queries are answered as 'yes' if the number of detected bounding boxes is equal to the number mentioned in the question. For evaluation queries measuring text rendering skill (*e.g.*, "Do the letters say 'Contour'?"), an OCR module on detects text from images. The queries are answered as 'yes' if the target text is captured.

Overall, with the assistance of domain-specific models, we observe difference performance in the VQA-human alignment: slightly lower performance with OD (Tab. 17), and noticeable (yet not categorically substantial) bump with OCR (Tab. 18). The results are consistent with the findings in Cho et al. (2023b).

Table 15: VQA-Human correlation. VQA: proportion of correct answers; Human: 1-5 Likert. **Split by fine-grained semantic categories**, measured with Spearman's $\rho$ and Kendall's $\tau$. *ent*: entity, *rel*: relation, *att*: attribute. (Data: TIFA160; Categories: DSG-1k question types).

| VQA models | DSG-1k Question types | | | | | | | |
|---|---|---|---|---|---|---|---|---|
| | (ent) whole | (rel) spatial | (att) state | (ent) part | (global) | (att) color | (rel) action | (att) type |
| PaLI | 0.499/0.397 | 0.518/0.423 | 0.401/0.330 | 0.410/0.332 | 0.243/0.213 | 0.291/0.267 | 0.389/0.311 | 0.359/0.310 |
| | (att) count | (att) text rendering | (att) material | (att) shape | (att) style | (att) texture | (att) size | - |
| PaLI | 0.346/0.319 | - | - | - | - | - | - | - |

Table 16: T2I model VQA accuracy (SD v2.1 / Imagen* / MUSE*) (averaged over text-image items). **By fine-grained semantic categories**. Questions generated with DSG-PaLM2, answered with both PaLI and human. *ent*: entity, *rel*: relation, *att*: attribute.

| T2I models | DSG-1k Question types | | | | | | | |
|---|---|---|---|---|---|---|---|---|
| | (ent) whole | (rel) spatial | (att) state | (ent) part | (global) | (att) color | (rel) action | (att) type |
| **Answerer: PaLI** | | | | | | | | |
| SD v2.1 | 81.9 | 63.8 | 85.9 | 84.4 | **90.7** | 76.8 | 73.9 | 78.2 |
| Imagen* | **85.3** | **71.2** | 87.6 | 87.6 | **90.7** | 87.4 | **78.3** | **81.9** |
| MUSE* | 83.8 | 70.1 | **88.0** | **88.2** | 93.1 | 87.1 | 75.0 | 80.2 |
| **Answerer: Human** | | | | | | | | |
| SD v2.1 | 68.2 | 46.3 | 52.6 | 76.1 | 48.9 | 67.3 | 46.9 | 44.7 |
| Imagen* | **74.3** | **58.6** | **62.1** | 84.2 | **50.0** | 78.2 | **62.0** | **46.8** |
| MUSE* | 71.4 | 55.8 | 58.3 | 81.3 | 46.2 | **78.7** | 54.6 | 41.6 |
| | (att) count | (att) text rendering | (att) material | (att) shape | (att) style | (att) texture | (att) size | |
| **Answerer: PaLI** | | | | | | | | |
| SD v2.1 | 73.5 | 70.6 | 73.2 | 82.0 | 79.6 | 71.8 | 80.0 | |
| Imagen* | **80.1** | 70.6 | **81.4** | 86.0 | 77.6 | **94.9** | 85.7 | |
| MUSE* | 77.3 | **72.5** | 78.4 | **88.0** | **83.7** | 89.7 | **94.3** | |
| **Answerer: Human** | | | | | | | | |
| SD v2.1 | 44.9 | **5.1** | 40.7 | 45.3 | 14.6 | 47.4 | 53.2 | |
| Imagen* | **49.7** | 1.8 | 51.2 | **55.6** | **17.6** | 72.4 | **63.1** | |
| MUSE* | 47.9 | 3.0 | **51.5** | 46.1 | 15.7 | 56.9 | 62.2 | |

**OD for `counting`.** Across TIFA, VQ$^2$A, and PaLI, we compare direct VQA query with the alternative query strategy with OWLv2-base-patch16 (Minderer et al., 2023) on the DSG-1k split dedicated to `counting` – CountBench (Paiss et al., 2023). The result is summarized in Tab. 17.

| VQA models | direct VQA query (ours) | +OD for counting |
|---|---|---|
| mPLUG-large | 65.7 | 64.1 (-1.6) |
| Instruct-BLIP | 60.2 | 58.9 (-1.3) |
| PaLI | 66.1 | 64.9 (-1.2) |

Table 17: VQA-human matching accuracy of VQA models with direct VQA query vs. +OD (w/ VPEval). Data split: CountBench.

**OCR for `text rendering`.** Similar to the experiment with OD, on the DSG-1k split on `text rendering` – DrawText (Liu et al., 2023), we again compare direct VQA query with the alternative query strategy with EasyOCR (Jaided AI, 2023) integrated per VPEval. The result is summarized in Tab. 18.

| VQA models | direct VQA query (ours) | +OCR for text rendering |
|---|---|---|
| mPLUG-large | 60.3 | 64.5 (+4.2) |
| Instruct-BLIP | 57.0 | 63.5 (+6.5) |
| PaLI | 60.3 | 66.8 (+6.5) |

Table 18: VQA-human matching accuracy of VQA models with direct VQA query vs. +OCR (w/ VPEval). Data Split: DrawText.

## G    PSEUDOCODE OF T2I EVALUATION WITH DSG.

Algorithm 1 shows Python pseudocode demonstrating the T2I evaluation pipeline with DSG.

---

**Algorithm 1** Python pseudocode of T2I evaluation with DSG

```python
PROMPT_TUPLE = """Task: given input prompts,
describe each scene with skill-specific tuples ...
"""

PROMPT_DEPENDENCY = """Task: given input prompts and tuples,
describe the parent tuples of each tuple ...
"""

PROMPT_QUESTION = """Task: given input prompts and skill-specific tuples,
re-write tuple each in natural language question ...
"""

def generate_dsg(text, LLM):
    """generate DSG (tuples, dependency, and questions) from text"""
    # 1) generate atomic semantic tuples
    # output: dictionary of {tuple id: semantic tuple}
    id2tuples = LLM(text, PROMPT_TUPLE)
    # 2) generate dependency graph from the tuples
    # output: dictionary of {tuple id: ids of parent tuples}
    id2dependency = LLM(text, id2tuples, PROMPT_DEPENDENCY)
    # 3) generate questions from the tuples
    # output: dictionary of {tuple id: ids of generated questions}
    id2questions = LLM(text, id2tuples, PROMPT_QUESTION)
    return id2tuples, id2dependency, id2questions

def evaluate_image_dsg(text, generated_image, VQA, LLM):
    """evaluate a generated image with DSG"""
    # 1) generate DSG from text
    id2tuples, id2dependency, id2questions = generate_dsg(text, LLM)
    # 2) answer questions with the generated image
    id2scores = {}
    for id, question in id2questions.items():
        answer = VQA(generated_image, question)
        id2scores[id] = float(answer == 'yes')
    # 3) zero-out scores from invalid questions
    for id, parent_ids in id2dependency.items():
        # zero-out scores if parent questions are answered 'no'
        any_parent_answered_no = False
        for parent_id in parent_ids:
            if id2scores[parent_id] == 0:
                any_parent_answered_no = True
                break
        if any_parent_answered_no:
            id2scores[id] = 0
    # 4) calculate the final score by averaging
    average_score = sum(id2scores.values()) / len(id2scores)
    return average_score
```

---

## H    HUMAN EVALUATION SETUP

Fig. 11 and Fig. 12 exemplify the data collection UI used in our human answer/rating collection.

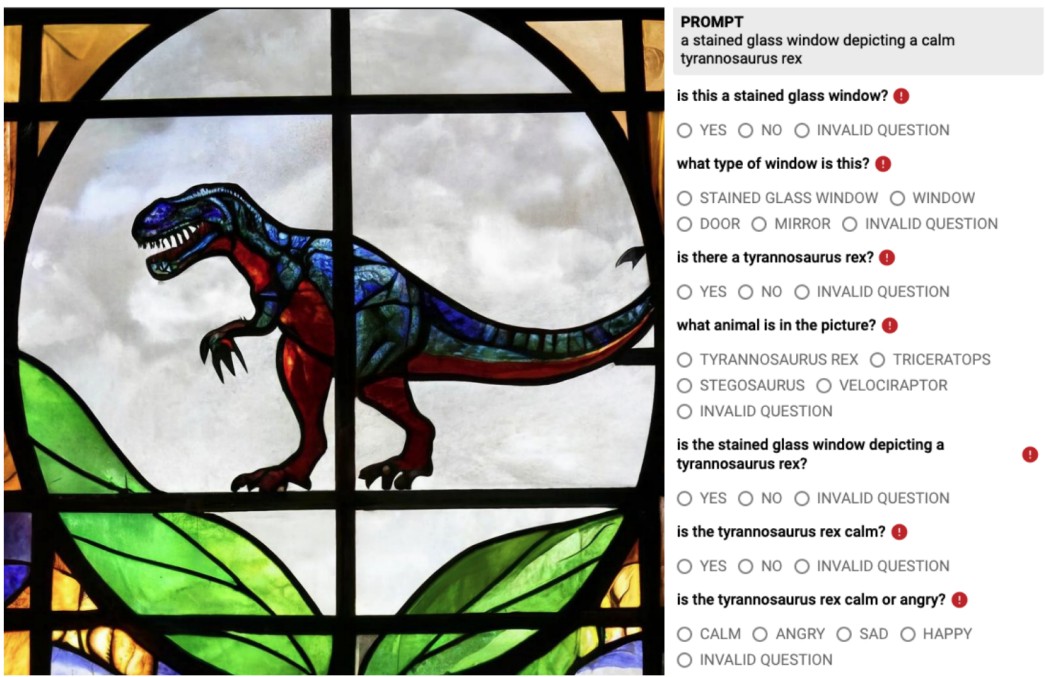

Figure 11: UI for data collection in per-question human judgment annotation.

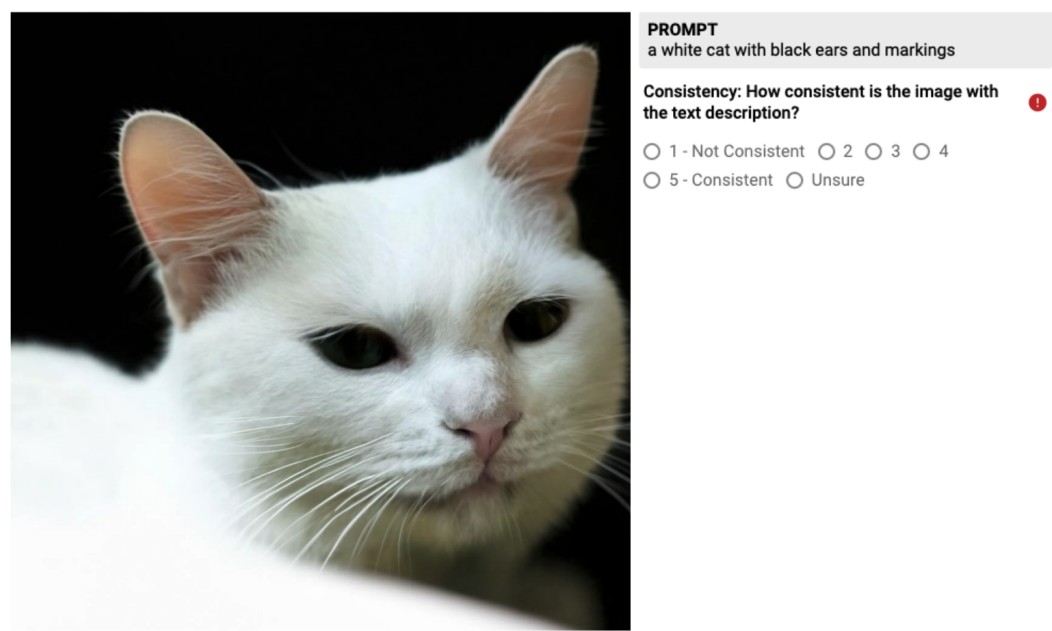

Figure 12: UI for data collection in per-item human judgment annotation (1-5 Likert consistency rating).

