# OpenReview forum: "Davidsonian Scene Graph: Improving Reliability in Fine-grained Evaluation for Text-to-Image Generation"
_ICLR.cc/2024/Conference — ICLR 2024 poster_

### Official Review · Reviewer_p5Xv · 2023-10-22

**Soundness:** 2 fair
**Presentation:** 3 good
**Contribution:** 3 good
**Rating:** 5
**Confidence:** 3

**Summary:**

This paper studies the problem of fine-grained evaluation of text-to-image (T2I) alignment. Following a line of recent works, this paper formulates T2I alignment evaluation as visual question answering (VQA), which involves question generation from the text prompt using LLMs and question answering using VQA models (referred to as the QG/A framework).

The main contributions of this paper are:
* It identifies the issues of existing QG/A methods regarding the question generation step and proposes four properties that the generated questions should satisfy.
* It proposes a QG method that constructs the questions of a given text prompt as a Davidsonian Scene Graph (DSG). DSG considers the dependency between questions and is designed to satisfy the four properties.
* It collects a test set of 1060 prompts, which covers different challenges, semantic categories and writing styles, together with Likert-scale T2I alignment rating by humans.
* The experiments demonstrate that DSG, when combined with different VQA models, achieves a higher correlation with human evaluation compared with existing QG/A frameworks.
* It further reveals two challenges of QG/A frameworks for T2I evaluation: (1) Some question categories (e.g., shape, style and text rendering) are beyond the capability of current SOTA VQA models to evaluate. (2) For questions that involve “subjectivity” and “domain knowledge”, agreement is hard to achieve even between humans.

**Strengths:**

* This work reveals the issues of existing QG/A methods in terms of the QG step, which is well-motivated and the proposed four desired properties of QG are reasonable.
* The experiments demonstrate that the proposed DSG achieves solid improvement over existing QG/A frameworks in terms of correlation with human evaluation, which advances the reliability of automatic T2I alignment evaluation.
* This work is transparent about its limitations, shedding light on directions for future studies to work on.
* The paper is well-written and easy to follow.

**Weaknesses:**

### The design of DSG and DSG-1k
* It is not well-explained which part of DSG is designed to address the **Atomic Question** and **Full Semantic Coverage** properties. Why the QG methods of TIFA and VQ^A cannot achieve these two properties?
* Some categories are under-represented in the DSG-1k dataset. According to Table 9, six categories have less than 30 examples and VQA-Human correlation is not computed for these categories.

### Evaluation
* There is no ablation study showing which design choice in DSG contributes to each of the four properties.
* It is unclear whether DSG correlates better with humans, compared with TIFA and VQ^A, in each fine-grained category.
* The relationship between the claimed desired question properties and the final VQA-Human correlation is not well demonstrated. In other words, the relationship between Precision/Recall/Atomicity/Uniqueness in Table 2 and Spearman/Kendall correlation in Table 3 is unclear.
* There is no comparison with TIFA and VQ^2A in terms of precision and recall in Table 2.

**Questions:**

* It is stated on page 4 that DSG only involve binary yes/no questions. Why are there multi-choice questions in the human annotation UI in Figure 10?

---

> ### Author Response · Authors · 2023-11-19
> **Author response to reviewer p5Xv (Part 1)**
>
> We thank reviewer p5Xv for the valuable comments on the design and evaluation of DSG / DSG-1k. We address/clarify all the points and report the revision we applied to our submission.
>
> > **W1. It is not well-explained which part of DSG is designed to address the Atomic Question and Full Semantic Coverage properties. Why the QG methods of TIFA and VQ^A cannot achieve these two properties?**
>
> Thanks for the great suggestion — we made a revision to explain the point more clearly (Sec. 3, the paragraph above the subsection on DSG-1k dataset).
>
> In gist, to the question “why the QG methods of TIFA and VQ^2 cannot achieve atomicity and semantic coverage?”, the short answer is: because they do not have a structured intermediary representation in the question generation process (like DSG’s semantic tuples) to a) define the semantic coverage for a prompt, and b) thereby allowing for the verification of atomicity.
>
> To expand the answer in detail, DSG uses semantic tuples (e.g., attribute(bike, blue), entity(bike), relation(bike, wall, next to)) as an intermediary representation in question generation: prompt -> semantic tuples -> questions. As such, leveraging the structuredness of the semantic tuple intermediary representation, we can then clearly define "atomicity" and "semantic coverage" and thereby modulate DSG's behavior to generate questions accordingly:
>
> - Atomicity: an atomic question corresponds to only a single semantic tuple. For example, "is there a bike?" maps to one semantic tuple "entity(bike)", and is atomic; "is there a blue bike?" maps to two semantic tuples "entity(bike)" and "attribute(bike, blue)", and is thus NOT atomic.
> - Semantic coverage: the full semantics of a prompt is represented by a set of unique semantic tuples (no duplicates), such that no entities, attributes, relations, or global descriptions can be added.
>
> TIFA and VQ^2, on the other hand, generate questions directly, thus there is no means available to encourage atomicity and semantic coverage (i.e., with a direct prompt-to-questions configuration w/o semantic tuples intermediary). In our human evaluation of TIFA and VQ^2 questions (Table 2), we indeed find large numbers of non-atomic questions and violations of semantic coverage (with duplication and missing entities/attributes/relations).
>
>
> > **W2. Some categories are under-represented in the DSG-1k dataset. According to Table 9, six categories have less than 30 examples and VQA-Human correlation is not computed for these categories.**
>
> We apologize for the confusion introduced here. Please allow us to clarify (also added clarification in the caption the table referred to – Table 13 (Table 9 in the original PDF)):
>
> Specifically, Table 13 of the revised PDF (Table 9 in the original PDF) examines questions generated from **TIFA160 prompts** (160 in total), where we used DSG-1k’s ‘question types’ to present specific scores,  rather than from the whole DSG-1k prompts. TIFA160 is a relatively small sample, so they do not reflect some types of questions such as text rendering. In Table 14 of the revised PDF (Table 10 in the original PDF), we illustrated three T2I models’ VQA accuracy in the full range of DSG-1k’s semantic categories.
>
>
>
> > **W3. There is no ablation study showing which design choice in DSG contributes to each of the four properties.**
>
> As a similar point has also been raised by others, to avoid duplication, we reply in the common response. Please refer to the common response. Thanks.
>
> > **W4. It is unclear whether DSG correlates better with humans, compared with TIFA and VQ^A, in each fine-grained category.**
>
> We agree that a by-category human correlation study is a valuable addition. Following the suggestion, we conducted an additional experiment using TIFA160 prompts, by calculating correlation Likert 1-5 scores with each of their 4 data splits:
> - COCO: general purpose;
> - PaintSkills: focusing on spatial relations;
> - PartiPrompts: focusing on attributional failure cases (e.g., color, counting, etc.);
> - DrawBench: focusing on simple compositionality.
>
> As shown in the following table, DSG produces higher human correlation scores on 3 out of 4 splits, as well as the best overall performance. Even in the only data split it does not come on top (i.e., DrawBench), the DSG’s correlation is very close to the top-performing TIFA’s. We incorporated the new experiment result into Appendix D.2 in our revised PDF, thanks for the suggestion!
>
> |        | COCO     | PaintSkills | PartiPrompts | DrawBench | Overall  |
> |---|---|---|---|---|---|
> | TIFA   | 0.30/0.22| 0.56/0.40  | 0.49/0.37    | **0.59/0.46** | 0.43/0.32|
> | VQ^2A  | 0.17/0.13| 0.25/0.17  | 0.28/0.21    | 0.45/0.37 | 0.21/0.16|
> | DSG    | **0.53/0.42** | **0.64/0.53**  | **0.61/0.49**    | 0.56/0.44 | **0.57/0.46** |
>
> (continued below)

---

> ### Author Response · Authors · 2023-11-19
> **Author response to reviewer p5Xv (Part 2)**
>
> > **W5. The relationship between the claimed desired question properties and the final VQA-Human correlation is not well demonstrated. In other words, the relationship between Precision/Recall/Atomicity/Uniqueness in Table 2 and Spearman/Kendall correlation in Table 3 is unclear.**
>
> For context, we would like to clarify that the Precision/Recall/Atomicity/Uniqueness validation (Table 2) is intended to check the quality of the generated questions on the extent to which they match human judgment, i.e., “are these the right validation questions to ask to check T2I alignment?”, and importantly, achieving strong human correlation in the answers while maintaining the high quality in question generation.
>
> A natural doubt that we expect may arise is: without having the atomicity, semantic coverage, etc. quality we use to evaluate generated questions, TIFA and VQ^2 can still achieve decent human correlation, then what’s the point in having these quality metrics?
>
> We believe the answer lies with **interpretability** and **reliable diagnostics**, especially on the instance/question-level — for any individual question, we strive to achieve an unambiguous understanding of human judgments. For example, a response to a non-atomic question “Is there a blue bike parked next to a door?” would be hard to interpret — if the answer to this question is ‘no’, then is it that the ‘bike’ is not generated properly? or is it that the ‘location of the bike’ is wrong?
>
> Therefore, to summarize:
> - With the Precision/Recall/Atomicity/Uniqueness validation, we show that DSG produces high-quality questions per human intuition. This ensures interpretability and reliable diagnostics;
> - With the human correlation experiments, we show that DSG results at the same time correlate well (and better than previous work) with human judgments.
>
>
> > **W6. There is no comparison with TIFA and VQ^2A in terms of precision and recall in Table 2.**
>
> We agree it is possible to evaluate TIFA and VQ^2 by eliciting human judgments on precision and recall, however in practice, there exists a failure mode that renders the evaluation not implementable (indicated by their low atomicity in Table 2) — the issue is rooted in how TIFA/VQ^2 and DSG are designed – we clarify as follows:
>
> Note that, due to the absence of the concept of atomicity, we frequently observe TIFA and VQ^2 questions that cover two or more semantic details (e.g. “is there a blue bike next to the door?” covers the entity “bike”, the color of it, and its relative position to another entity “door”). As we argue in the Sec. 1 introduction (the A-B-C-D bullet points on reliability), “loaded questions” like these do not lend themselves well to acquiring unambiguous answers (either by a VQA model or a human rater).
>
> If we disregard the reliability concern to look at precision/recall alone, loaded questions can score high precision/recall, by simply adding a question mark to the original prompt to form a question (e.g., “A blue motorcycle parked by paint chipped doors.” $\rightarrow$ “A blue motorcycle parked by paint chipped doors?”). This would render the precision/recall validation uninformative.
>
> Our DSG, on the other hand, uses semantic tuples as a structured intermediary semantic representation to handle atomicity – this wards it off the “loaded question” failure mode mentioned above in the precision/recall validation (e.g., if a “loaded question” that maps to more than 1 semantic tuple is produced, it is considered a wrong question).
>
> > **Q1. It is stated on page 4 that DSG only involves binary yes/no questions. Why are there multi-choice questions in the human annotation UI in Figure 10?**
>
> It is true that DSG only involves binary yes/no questions. The questions inside Figure 10 are generated by TIFA, where multiple-choice questions are permitted. We use the same human annotation UI for questions generated by DSG and VQ2A, where we provide yes/no answer choices.

---

> > ### Comment · Reviewer_p5Xv · 2023-11-22
> > **Response to Author Rebuttal**
> >
> > Thank you for the detailed reply. Most of my concerns have been addressed. I still have some further suggestions and comments:
> > - It seems that the current paper lacks information on DSG-1k's data distribution over the categories (whole, spatial, color, count, etc). It is important to show that each category contains a sufficient number of examples to ensure statistically meaningful evaluation.
> > - About W3: I'm afraid that there is a misunderstanding in your response. What I expect is ablating components in DSG (e.g., DSG w/o dep) and observing the impact on the four properties (i.e., Precision, Recall, Atomicity and Uniqueness in Table 2), instead of (+atomicity) and (+atomicity, +uniqueness), etc.
> > - About W6: I agree that it is meaningless to compare precision/recall in the absence of atomicity. It would be better to explain this in the paper.

---

> > > ### Author Response · Authors · 2023-11-22
> > > **Response to the additional questions of Reviewer p5XV**
> > >
> > > We sincerely appreciate you taking the time to give your additional comments.
> > >
> > > > **It seems that the current paper lacks information on DSG-1k's data distribution over the categories (whole, spatial, color, count, etc). It is important to show that each category contains a sufficient number of examples to ensure statistically meaningful evaluation.**
> > >
> > > We agree that it is important to have a sufficient number of questions to conduct question category-specific analysis. Following your suggestion, in Appendix B in the revised PDF, we provide two additional tables showing the number of questions per broad semantic categories (e.g., entity, attribute, relation, global) and detailed semantic categories (e.g., whole, spatial, color, count). The tables show that DSG-1k questions have enough category-specific questions to ensure statistically meaningful evaluation.
> > >
> > > > **About W3: I'm afraid that there is a misunderstanding in your response. What I expect is ablating components in DSG (e.g., DSG w/o dep) and observing the impact on the four properties (i.e., Precision, Recall, Atomicity and Uniqueness in Table 2), instead of (+atomicity) and (+atomicity, +uniqueness), etc.**
> > >
> > > We appreciate the further clarification on your point. DSG is composed of three components: the semantic tuples, the questions, and the dependencies. As we understand, there are two components to ablate you are referring to - semantic tuples and dependencies. **However**, it is crucial for us to point out the cost of losing semantic tuples and dependencies.
> > >
> > > Semantic tuples mark the semantic types of the questions which is crucial for our question category-specific analysis. Without the semantic tuples, we cannot provide the comprehensive category-specific analysis like in Tables 4, 5, 14, etc. Moreover, we would like to point out that the questions shown to the LM in-context examples are derived from the semantic tuples, which are designed to be atomic, unique, and covering the full semantics of the prompts.
> > >
> > > Regarding dependency ablation, as dependency is generated independently to questions, dependency does not affect the questions evaluation criteria (Precision, Recall, Atomicity, and Uniqueness in Table 2). Moreover, we already comprehensively report the usefulness of dependency in Appendix D Table 8: dependencies are shown to be crucial in improving VQA-Human matching accuracy, by preventing the evaluation systems from asking invalid VQA queries (e.g., “is there a bike?” VQA: NO  $\rightarrow$  “is the bike blue?”).
> > >
> > > Finally, we would like to emphasize that DSG shows significant improvement over the QG/A frameworks (i.e., TIFA&VQ^2) that do not use our core components (semantic tuples and dependencies), in uniqueness and atomicity (in Table 2). The semantic tuples contribute to 3 of the 4 reliability dimensions we define in the introduction – atomicity, uniqueness, and full semantic coverage (by precision and recall). The dependency contributes to the last reliability property (i.e. valid dependency). Our experiments show the proposed configuration of DSG improves VQA-Human matching accuracy **while** maintaining the 4 properties.
> > >
> > >
> > > > **About W6: I agree that it is meaningless to compare precision/recall in the absence of atomicity. It would be better to explain this in the paper.**
> > >
> > > Agreed. Following your suggestion, we have added the explanation in Appendix E of the revised PDF.
> > >
> > >
> > > =========================================================
> > >
> > > We hope our responses have addressed your additional comments. It would be great if you could increase the rating, as you mentioned that ‘most of my concerns have been addressed’. Thanks!

---

> > > > ### Comment · Reviewer_p5Xv · 2023-11-23
> > > > **Response to Author Rebuttal**
> > > >
> > > > Thanks for the response. I still have a further question related to W1 and W3: TIFA also generates intermediary representation (extracting elements) before generating the questions. As such, I'm interested in understanding whether the better performance of DSG over TIFA stems from (1) the utilization of semantic tuples as a more effective representation than individual elements, (2) the incorporation of better in-context examples, or any other factors?
> > > >
> > > > I will consider raising the rating based on the response to this question.

---

> ### Author Response · Authors · 2023-11-23
> **Response to the additional questions**
>
> Thanks for your additional comments!
>
> > I'm interested in understanding whether the better performance of DSG over TIFA stems from (1) the utilization of semantic tuples as a more effective representation than individual elements, (2) the incorporation of better in-context examples, or any other factors?
>
> We would like to point out that the most major difference between DSG and TIFA is that DSG has a dependency structure across each entity/attribute/relation, while the extracted elements in TIFA does not.
>
> Consider the caption "A blue motorcycle". There are two questions, "is there a motorcycle?" and "is the motorcycle blue?". If there is no motorcycle in the image, then it is just unreasonable to ask "is the motorcycle blue?". The design of DSG makes sure that, if the entity "motorcycle" is not there, then the attribute "blue" is considered missing. In TIFA, there is no such guarantee, and we observe that the VQA models sometimes just hallucinate answers even if the question is not reasonable, which leads to bias in evaluation.
>
> As such, the dependency structure of DSG is critical, and it is where DSG is making a difference compared with TIFA and VQ2. Besides, the improvement on precision, recall, atomicity, uniqueness are side benefits we observed, and the points you listed are possible explanations for these side benefits.
>
> We hope our response has addressed your concerns. As it is the end of the discussion period, we do not have time to write more. It would be great if you increase the rating. Thanks!

---

### Official Review · Reviewer_A5bo · 2023-10-29

**Soundness:** 3 good
**Presentation:** 3 good
**Contribution:** 3 good
**Rating:** 8
**Confidence:** 3

**Summary:**

This paper focuses on text-to-image generation evaluation by asking text-related questions and checking whether a VQA model can answer it given the generated image. It suggests previous QG/A frameworks usually ask ambiguous, duplicated, and invalid questions. The paper parses the input text input atomic entity/attribute/relation tuples, translates each tuple into questions, and obtains their dependencies through an LLM. The experimental results show the generated questions are unique, have valid entailment, and query atomic semantics. The paper lastly constructs a dataset with the proposed methods.

**Strengths:**

The paper has strong motivation. It first analyses the drawbacks of previous QG/A methods, then proposes some principles for the generated questions, and lastly introduces a three-step prompting method to resolve it.

The experimental results are strong enough to support the effectiveness of the proposed method in the proposed uniqueness, valid dependency, and human alignment.

The paper is well-written, and the related work is sufficient.

**Weaknesses:**

The VQA model is not good enough and hinders the final alignment to humans in T2I evaluation.

**Questions:**

Can a better VQA model lead to better alignment in Tables 3, 4, and 6?

---

> ### Author Response · Authors · 2023-11-19
> **Author response to reviewer A5bo**
>
> We thank reviewer A5bo for the feedback and comments.
>
> > **W1. The VQA model is not good enough and hinders the final alignment to humans in T2I evaluation.**
>
> As a similar point has also been raised by others, to avoid duplication, we reply in the common response. Please refer to the common response. Thanks.
>
> > **Q1. Can a better VQA model lead to better alignment in Tables 3, 4, and 6?**
>
> Yes. This can be found in the results we already report in the paper. Please allow us to clarify:
>
> Empirically, based on the result in Table 5 (per-question VQA-Human matching accuracy), we see a common trend in Tables 3, 4, and 6 — if a VQA model’s result correlates better with human Likert scores, the model also tends to have a higher matching accuracy on the per-question level.

---

### Official Review · Reviewer_tDEJ · 2023-10-30

**Soundness:** 2 fair
**Presentation:** 2 fair
**Contribution:** 2 fair
**Rating:** 5
**Confidence:** 5

**Summary:**

The paper addresses the task of evaluating text-to-image models, specifically focusing on the question generation and answering method. This method automatically generate questions and answers from the prompt, and the faithfulness of the image is assessed based on the consistency of the answers from both prompt and visual question answering models. The authors identify and tackle key reliability challenges in this approach, including the quality of generated questions and the consistency of visual question answering.
To overcome these challenges, the authors introduce the Davidsonian Scene Graph (DSG), which produces atomic and unique questions organized in dependency graphs, to ensure questions cover semantic of the prompt and that answers are consistent.
The paper provides experimental results and human evaluations, demonstrating that DSG addresses the reliability challenges mentioned earlier. Additionally, the authors introduce DSG-1k, an open-sourced evaluation benchmark with 1,060 prompts covering a wide range of fine-grained semantic categories.

**Strengths:**

This paper introduces Davidsonian Scene Graph (DSG) to improve the faithfulness of the text-to-image evaluation. Compared to previous QG/A methods, this framework generates  atomic questions with full semantic coverage and valid question dependency. The authors implement QG step as a Directed Acyclic Graph (DAG) where the nodes represent the unique questions and they explicitly model semantic dependencies with directed edges. Additionally, they collect a fine-grained human-annotated benchmark called DSG-1k including 1,060 diverse prompts with a balanced distribution to facilitate research in this area.

**Weaknesses:**

1.	This paper improves the existing QG/A methods with Davidsonian Scene graph, which is generated based on LLMs. The approach of the work could be enriched with more details and techniques.

2.	Ablation on separate steps of DSG should be presented. For example, without establishing dependencies, measure the changes of consistency between VQA score and the human 1-5 Likert Scores.

3.	Apart from TIFA and VQ2A, more methods could be compared, including CLIPScore and caption based approaches.

**Questions:**

1.	In 4.1, to validate the question dependencies, authors evaluate manually on 30 samples and automatically on the full TIFA160. However, the consistency of manual and automatic evaluation is not presented.

2.	The comparison of runtime among different methods should be added.

3.	In table 3, for Instruct-BLIP, the Spearman’s ρ of DSG is lower than that of TIFA, authors should explain this phenomenon briefly.

---

> ### Author Response · Authors · 2023-11-19
> **Author response to reviewer tDEJ (Part 1)**
>
> We thank reviewer tDEJ for the insightful and detailed comments. We address/clarify all the weaknesses they bring to our attention and report the revision we applied to our submission.
>
> > **W1. This paper improves the existing QG/A methods with Davidsonian Scene graph, which is generated based on LLMs. The approach of the work could be enriched with more details and techniques.**
>
> As a similar point has also been raised by others, to avoid duplication, we reply in the common response. Please refer to the common response. Thanks.
>
> > **W2. Ablation on separate steps of DSG should be presented. For example, without establishing dependencies, measure the changes of consistency between VQA score and the human 1-5 Likert Scores.**
>
> As a similar point has also been raised by others, to avoid duplication, we reply in the common response. Please refer to the common response. Thanks.
>
> > **W3. Apart from TIFA and VQ2A, more methods could be compared, including CLIPScore and caption based approaches.**
>
> Please note that we mentioned “For human correlation, Hu et al. (2023) for example achieves 47.2 Kendall’s τ in a sizable correlation analysis with human 1-5 Likert judgments, with CLIPScore τ = 23.1.’’,
> For completeness, we also calculated and included correlation results on the 5 single-summary metrics considered in Hu et al. (2023), i.e., BLEU-4, ROUGE-L, METEOR, SPICE, (with BLIP-2 Captioning) and CLIPScore (with CLIP ViT-B/32) in Appendix E in the revised PDF.
>
>
> |  | Spearman's ρ | Kendall's τ |
> |---|---|---|
> | **Captioning  (w/ BLIP-2)** |
> | BLEU-4 | 26.1 | 18.3 |
> | ROUGE-L | 34.2 | 23.9 |
> | METEOR | 37.9 | 26.9 |
> | SPICE | 32.9 | 23.6 |
> | **Cosine similarity (CLIP ViT-B/32)** |
> | CLIPScore | 27.6 | 19.1 |
> | **QG/A frameworks  (w/ PaLI)** |
> | VQ$^2$A | 20.7 | 15.7 |
> | TIFA | 43.1 | 32.3 |
> | DSG |  **57.0** | **45.8** |
>
> (NOTE: To ensure consistency, we did not directly report the result from Hu et al. (2023) Table 2. Our numbers are calculated with the same human 1-5 Likerts **we** collected – the same Likerts are used in calculating the correlation results in our Table 3. Further, also note that Hu et al. (2023) calculated human Likerts based (by rules, see their Appendix C.1) on raters’ answers to the GPT-3 generated questions, whereas we collected them independently with a single 1-5 consistency question, please see our Figure 12.)
>
> Further, we set focus on fine-grained QG/A as an overall more informative / diagnostic framework than single-summary metrics (represented by CLIPScore), and intend to show that, not only does QG/A work well on the coarse-grained model-level in evaluating performance (as TIFA/VQ^2 show in their comparison to a range of single-summary methods), we can improve it further (with DSG) to provide reliable fine-grained diagnostics.
>
> (continued below)

---

> ### Author Response · Authors · 2023-11-19
> **Author response to reviewer tDEJ (Part 2)**
>
> > **Q1. In 4.1, to validate the question dependencies, authors evaluate manually on 30 samples and automatically on the full TIFA160. However, the consistency of manual and automatic evaluation is not presented.**
>
> Following the suggestion, we additionally report the the consistency of 30 samples of manual v.s automatic evaluation — dependency valid ratio of DSG is 100% in manual evaluation (30 samples), and 99% in automatic evaluation (full TIFA160). In the revised PDF, we added a sentence to present the result more clearly.
>
> > **Q2. The comparison of runtime among different methods should be added.**
>
> We compare the runtime of TIFA and our DSG. For apple-to-apple comparison, we use the same LLM (gpt-3.5-turbo-16k) and VQA model (mPLUG-large). On a single A6000 GPU, we measure the time for obtaining the final T2I alignment score for 20 image-text pairs. We use 20 prompts sampled from TIFA-160 prompts and images generated by Stable Diffusion v2.1.
>
> As in the following result, we find DSG takes a shorter time than TIFA, primarily because TIFA tends to generate more questions than DSG, increasing the inference time of VQA models.
>
> * TIFA: 481s for 20 image-text pairs (24s per image-text pair)
> * DSG: 332s for 20 image-text pairs (16s per image-text pair)
>
> Please note that the cost for many image-text pairs could be saved in many ways: e.g., batch processing (separate QG and QA stages, processing multiple images), caching (re-using image features when a VQA model answers to questions paired with the same image), using smaller models (through quantization, distillation, compression), etc.
>
> > **Q3. In Table 3, for Instruct-BLIP, the Spearman’s ρ of DSG is lower than that of TIFA, authors should explain this phenomenon briefly.**
>
> While Spearman's rho shows "Instruct-BLIP & TIFA" > "Instruct-BLIP & DSG", Kendall's Tau shows "Instruct-BLIP & TIFA" < "Instruct-BLIP & DSG". There are some cases with tied ranks, where Kendall’s Tau correlation is more reliable than Spearman’s rho correlation when handling data with tied ranks [1,2].
>
> [1] Taylor, Wilson L. “Correcting the Average Rank Correlation Coefficient for Ties in Rankings.” Journal of the American Statistical Association, vol. 59, no. 307, 1964, pp. 872–76. JSTOR, https://doi.org/10.2307/2283105. Accessed 19 Nov. 2023.
>
> [2] Marie-Therese Puth, Markus Neuhäuser, Graeme D. Ruxton, “Effective use of Spearman's and Kendall's correlation coefficients for association between two measured traits”, Animal Behaviour, Volume 102, 2015, Pages 77-84, ISSN 0003-3472, https://doi.org/10.1016/j.anbehav.2015.01.010.

---

### Author Response · Authors · 2023-11-19
**Common response (Part 1)**

We thank the reviewers tDEJ, A5bo, and p5Xv for their valuable comments.

Our paper presents DSG, a linguistically motivated novel text-to-image (T2I) alignment evaluation method that addresses the critical reliability issues existing in the line of work on fine-grained T2I evaluation. Further, we contribute a benchmark dataset DSG-1k which is balanced in a rich set of semantic categories, and is intended to thereby provide informative diagnostics for understanding the behavior of T2I models.

We are glad that the reviewers recognize our strengths:

- reveals the issue of existing QG/A methods (p5Xv)
- strong motivation/well-motivated  (A5bo / p5Xv)
- proposed method can generate atomic questions with full semantic coverage and valid question dependency (tDEJ)
- improve the faithfulness/reliability of text-to-image evaluation (tDEJ / p5Xv)
- introducing a fine-grained human-annotated benchmark DSG-1k to facilitate research (tDEJ)
- strong experiments demonstrate the DSG achieves solid improvements over existing QG/A frameworks (A5bo / p5Xv)
- transparent about limitations, shedding light on directions on future studies (p5Xv)
- sufficient related work (A5bo)
- well written and easy to follow (A5bo / p5Xv)

We address the queries from each reviewer in separate responses and have incorporated feedback in our revised PDF. Below, we answer the questions commonly asked by some reviewers.

(continued below)

---

> ### Author Response · Authors · 2023-11-19
> **Common response (Part 2)**
>
> ### tDEJ W1 & A5bo W1- enriched techniques & improvements for DSG
>
> **Both reviewers asked if the VQA module can be improved or enriched for stronger performance.** Inspired by VPEval [1] which adopts different evaluation models for different evaluation queries, for the challenging text rendering and counting skills, we experiment with using an expert module for specific skill (e.g., OD for counting and OCR for text rendering), and compare the results in Metric vs. Human answer match accuracy.
>
> For text rendering, we applied EasyOCR [2] for the questions measuring text rendering capability (e.g., “Do the letters say ‘Contour’?”) instead of the direct VQA method, on the DrawText split in DSG-1k. Specifically, we run OCR on images and check whether the target text is captured. As shown in the following table, we find the integration of OCR improved the Metric vs. Human answer match accuracy on the DrawText split with a strong margin.
>
> | VQA models     | VQA for all questions | + OCR for text rendering |
> |---|----|---|
> | mPLUG-large    | 60.3             | 64.5 (+4.2) |
> | Instruct-BLIP  | 57.0             | 63.5 (+6.5) |
> | PaLI           | 60.3             | 66.8 (+6.5) |
>
> For counting, we applied OWLv2 [3], a recent state-of-the-art open-vocab object detection model, for the questions measuring counting capability (e.g., "Are there five Halloween banners?”), instead of the direct VQA method, on the CountBench split in DSG-1k. Specifically, we detect bounding boxes with the target object name mentioned in the question and check whether the number of detected bounding boxes is equal to the number mentioned in the question. As shown in the following table, unlike OCR model in text rendering, the OWLv2 OD model didn’t improve the Metric vs. Human answer match accuracy on the counting skill.
>
> | VQA models     | VQA for all questions | + OD for counting |
> |---|----|---|
> | mPLUG-large    | 65.7                    | 64.1 (-1.6) |
> | Instruct-BLIP  | 60.2                    | 58.9 (-1.3) |
> | PaLI           | 66.1                    | 64.9 (-1.2) |
>
> In summary, we thank the reviewer again for the suggestion, and believe the inclusion of the result brings additional value in the understanding of the technique choice in handling the QA validation of the QG/A approach. We included the additional results in the Appendix H. Please also note that our DSG questions are labeled with the type of challenge that they probe (e.g.,  counting, text rendering), which makes them especially suited to be automatically routed to the area-specific techniques if they’re shown to perform better than the generalistic direct VQA approach.
>
> **In addition, reviewer tDEJ asked for further details on our DSG technique.** In regard to this, we included the pseudo-code for our full pipeline (question generation → question answering → calculating final score) in Appendix E in our original PDF, which exposes all the details of our algorithm. Furthermore, we attach the DSG code in the supplementary file.
>
> **Reviewer A5bo is curious whether a stronger VQA model would further increase the VQA-human alignment with the direct VQA query method.** Indeed, better VQA performance would help improve the quality of QG/A T2I evaluation in general. This is exactly the motivation for our extensive human evaluation in this work — identifying the semantic categories that can be automated with VQA models, and ones that remain in need of human judgments. We would like to emphasize that the question generation method (i.e. in the form of DSG) is what we consider the key innovation — on the back of the evidence illustrating the strong faithfulness of DSG-generated questions to human judgment (Table 2), we believe DSG is a QG approach that brings crucial and substantial improvement over previous work on the reliability of QG/A framework.
>
> ---------------------------------------------------------------------------------------
>
> [1] Cho et al., Visual Programming for Text-to-Image Generation and Evaluation, NeurIPS 2023.
>
> [2] Jaided AI,. Ready-to-use OCR with 80+ supported languages and all popular writing scripts including Latin, Chinese, Arabic, Devanagari, Cyrillic and etc., 2023.
>
> [3] Minderer et al., Scaling Open-Vocabulary Object Detection, 2023
>
> (continued below)

---

> ### Author Response · Authors · 2023-11-19
> **Common response (Part 3)**
>
> ### tDEJ W2 & p5Xv W3 - DSG ablation
>
> **Both reviewers suggested ablation on our DSG method from different angles.** We highly agree with the value in the additional experiments in this regard.
>
> **Reviewer tDEJ suggested that we look into the influence of DSG dependency annotations on the performance of VQA-human consistency.** Per the suggestion, we conducted additional experiments and reported our findings in the revision (Appendix D.1 of the revised PDF).
>
> We ablate the use of DSG dependencies in three conditions:
>
> a) DSG: Treating child questions that depend on parent questions with NO answers as NO. E.g. the child “is the bike blue?” depends on the parent “is there a bike?”, if the answer to the parent question is NO, then we assign NO as the answer to the child question.
> b) DSG (drop): Skipping the child questions that depend on parent questions with NO answers. E.g. With the running example above, if the answer to the parent question is NO, we do not factor the child question into the subsequent metric accuracy calculation.
> c) DSG (w/o dependencies): Ignoring all dependencies and treating all the questions equally.
>
> We then examined the effect in:
> - VQA-Human correlation. The Spearman & Kendall between the per-item VQA accuracy and human 1-5 Likert judgment.
> - VQA-Human matching accuracy. Per-question matching accuracy between VQA and human judgments. This is calculated as “number of questions where VQA answer and human answer agrees / number of total questions”
>
> For VQA-Human 1-5 Likert score correlation, we observe that the dependencies have little impact:
>
> | VQA models     | DSG        | DSG (drop) | DSG (w/o dep) |
> |---|---|---|---|
> | mPLUG-large    | 0.463/0.380| 0.462/0.370| 0.464/0.378   |
> | Instruct-BLIP  | 0.442/0.363| 0.437/0.358| 0.436/0.355   |
> | PaLI           | **0.570/0.458** | 0.561/0.451| 0.570/0.457   |
>
> However, on VQA-Human matching accuracy, using dependencies leads to a substantial improvement.
>
> | VQA models     | DSG    | DSG (drop) | DSG (w/o dep) |
> |---|---|---|---|
> | mPLUG-large    | 72.1   | 69.9       | 64.7          |
> | Instruct-BLIP  | 70.4   | 68.4       | 64.0          |
> | PaLI           | **73.8**   | 71.4       | 65.2          |
>
> Our findings can be summarized as:
> - While scores correlation to human 1-5 Likert has been the de-facto standard in (meta-)evaluation of QG/QA evaluation methods, and it is indicative of the rough tendency of the quality of the scores, we argue that (i) it is not fine-grained enough to tell apart the subtle differences between techniques, and (ii) it assumes that the human 1-5 Likert scores are the absolute source of truth, but it is likely that giving a holistic 1-5 classification of the alignment between a (potentially complex) prompt and an image is not a very well calibrated task.
> - We believe the per-question VQA-Human matching accuracy is much more effective on the fine-grained level for understanding exactly where VQA-Human match/mismatches reside, and is thereby more informative for model development as a diagnostic tool. DSG dependencies are empirically very valuable in improving VQA-Human matching accuracy.
>
> Beyond quantitative analysis, we firmly believe that taking into account the VQA model answer to “Is the car blue?” when there is no car in the image cannot bring any benefit to the evaluation results and thus we argue that those questions should not be asked.
>
> **Reviewer p5Xv suggested an ablation on DSG’s performance on the four properties we defined as the reliability criteria for a QG/A method — atomicity, uniqueness, dependency among questions, and full semantic coverage.** We venture to interpret the question as “why aren’t there a comparison study on different configuration combinations of DSG such as (+atomicity), (+atomicity, +uniqueness), (+atomicity, +uniqueness, +dependency) and so on, to ablatively probe each property.
>
> We agree that this would seem to serve as an informative dissection on the value of DSG in detail, however, a practical difficulty arises as a result. For example, say we have a condition where only atomicity is applied at the exclusion of all the other properties, we would need to, e.g., uniqueness: intentionally creating duplicates of questions; semantic coverage: intentionally avoiding covering the semantic details of a prompt in full. Each of the tasks falls on a continuous spectrum – how many duplications do we need to have to say the design is free of uniqueness? How many facts do we need to miss (percentage)?
>
> As such, we opted the following strategy instead:
> 1. Implementing DSG in a way that the generated questions should fulfill all the four desired properties as much as possible (**method description in Sec. 3**);
> 2. Validating the extent to which each property is fulfilled with both auto and human evaluation (**discussion in Sec. 4.1, result summary in Table 2**).

---

### Author Response · Authors · 2023-11-21
**A gentle reminder to check author responses**

Dear reviewers,

We again thank you for your valuable comments! We have responded to your comments, and would like to follow up to see if our responses address your concerns, or if you have any further questions. We would really appreciate the opportunity to discuss this further if our response has not already addressed your concerns.

Best,
Authors

---

### Meta-Review · Area_Chair_pU4B · 2023-12-06

**Metareview:**

This paper presents an approach to measure the quality of text to image generative models. The authors tackle fine-grained text alignment and measure it using question generation (from the text prompts) and answering using a VQA model (input generated questions and generated image). The paper identifies a set of requirements  that the automated question generation should follow and a linguistic constraint (Davidsonian Scene Graph) that ensures that these properties are satisfied.
They also collect a dataset of 1060 prompts and human likert ratings (DSG dataset).
Overall, the paper makes interesting contributions and observations  in evaluating these models which is both important and understudied.
During the review process, the reviewers raised several questions around ablations and design of the DSG dataset.
In particular, Rp5Xv raised important concerns about the dataset design, justification of design choices, and comparisons/relations to TIFA and VQ^2A. The author responses have resolved almost all the queries from the reviewers.
Given this resolution, and the contributions of this work I do not see any reason to not accept the paper.

**Justification For Why Not Higher Score:**

While the paper takes an important step in evaluating text to image models, it relies  heavily on automated question generation and linguistic structure of a single sentence text prompt. This approach may not easily work for multi-sentence long prompts that are becoming more common for the next generation of text-to-image models in the community. This will likely limit the impact of this particular paper.

**Justification For Why Not Lower Score:**

The paper is technically sound, addresses an important and understudied problem of evaluating text-to-image models, and has interesting contributions to this field.

---

### Decision · Program_Chairs · 2024-01-16

Accept (poster)